# Local and substrate-specific S-palmitoylation determines subcellular localization of Gαo

Gonzalo P. Solis [1✉], Arghavan Kazemzadeh[1], Laurence Abrami [2], Jana Valnohova[1], Cecilia Alvarez [3], F. Gisou van der Goot [2] & Vladimir L. Katanaev [1,4✉]

Peripheral membrane proteins (PMPs) associate with cellular membranes through post-translational modifications like S-palmitoylation. The Golgi apparatus is generally viewed as the transitory station where palmitoyl acyltransferases (PATs) modify PMPs, which are then transported to their ultimate destinations such as the plasma membrane (PM). However, little substrate specificity among the many PATs has been determined. Here we describe the inherent partitioning of Gαo – α-subunit of heterotrimeric Go proteins – to PM and Golgi, independent from Golgi-to-PM transport. A minimal code within Gαo N-terminus governs its compartmentalization and re-coding produces G protein versions with shifted localization. We establish the S-palmitoylation at the outer nuclear membrane assay ("SwissKASH") to probe substrate specificity of PATs in intact cells. With this assay, we show that PATs localizing to different membrane compartments display remarkable substrate selectivity, which is the basis for PMP compartmentalization. Our findings uncover a mechanism governing protein localization and establish the basis for innovative drug discovery.

[1] Translational Research Center in Oncohaematology, Department of Cell Physiology and Metabolism, Faculty of Medicine, University of Geneva, Geneva, Switzerland. [2] Global Health Institute, School of Life Sciences, EPFL, Lausanne, Switzerland. [3] Centro de Investigaciones en Bioquímica Clínica e Inmunología (CIBICI-CONICET). Departamento de Bioquímica Clínica, Facultad de Ciencias Químicas, Universidad Nacional de Córdoba, Córdoba, Argentina. [4] Institute of Life Sciences and Biomedicine, Far Eastern Federal University, Vladivostok, Russia. ✉email: gonzalo.solis@unige.ch; vladimir.katanaev@unige.ch

G protein-coupled receptors (GPCRs) and their immediate transducers—heterotrimeric G proteins—have been the subject of intensive scrutiny for decades, primarily due to their pivotal roles in innumerable physiological and pathological processes[1]. Heterotrimeric G proteins are composed of Gα, Gβ and Gγ subunits. The Gα subunit is loaded with either GDP or GTP; the Gβ and Gγ subunits form a constitutive heterodimer that reversibly binds to Gα. GPCRs directly interact with heterotrimeric G proteins on the cytosolic surface of the membrane. Upon activation, GPCRs act as exchange factors to enhance the release of GDP from Gα, leading to the binding of GTP and activation of the Gα subunit. Subsequently, activated Gα dissociates from the receptor and the Gβγ heterodimer, and the free subunits are competent to interact with downstream targets[2].

G protein activation via GPCRs has long been thought to occur exclusively at the plasma membrane (PM). Recently, however, considerable experimental evidence has accumulated supporting the notion that GPCRs can activate Gα subunits on the Golgi and other compartments[3]. Analogously, activation of the KDEL receptor (KDELR) by cargo from the endoplasmic reticulum (ER) was shown to trigger signal cascades via Gαs and Gαq/11, with KDELR acting as a non-canonical GPCR at the Golgi[4]. Our own work showed that KDELR also binds and activates monomeric Gαo, which in turn enhances the Golgi-to-PM trafficking via small Rab GTPases[5]. Therefore, the subcellular compartmentalization of Gα subunits appears to be of fundamental relevance for their functions. How Gα subunit compartmentalization is achieved and controlled, however, remains poorly understood.

Gα subunits are grouped into four subfamilies based on sequence and functional similarity: Gαs, Gαq/11, Gα12/13, and Gαo/i[2]. All Gα subunits bind to membranes via fatty acid modifications at the N-terminus, i.e., N-myristoylation and S-palmitoylation[6]. While the majority of Gα subunits are single-palmitoylated, Gαo and other members of the Gαo/i subfamily are dual lipidated. N-myristoylation occurs co-translationally and results in the attachment of a 14-carbon saturated fatty acid (myristate) to the N-terminal Gly via a stable amide bond[7]. S-palmitoylation occurs post-translationally and results in the attachment of a 16-carbon saturated fatty acid (palmitate) to a Cys residue through a reversible thioester linkage[8]. In vertebrates, N-myristoylation is catalyzed by two closely related N-myristoyltransferases (NMT1 and NMT2) whose substrate specificities have been intensively studied in recent years[9]. Intracellular S-palmitoylation is catalyzed by a zinc-finger Asp-His-His-Cys domain-containing (zDHHC) family of palmitoyl acyltransferases (PATs)[10]. There are up to 24 zDHHCs described in mammals; opposite to NMTs, their substrate specificities are far from being well understood, although a substantial advance has been made lately[11].

Previously, S-palmitoylation of peripheral membrane proteins (PMPs), including Gα subunits, was shown to occur exclusively at the Golgi, with the palmitoylated proteins subsequently transported to the PM[12]. In recent years, however, cumulative data has emerged indicating that some PMPs might undergo local S-palmitoylation on their target compartments, namely the PM or ER[13,14]. Experimental methods allowing visualization of S-palmitoylation in intact cells are highly demanded to properly address the issue of the locality of this crucial lipid modification.

Here, we define the critical parameters that govern Gαo N-myristolation, S-palmitoylation and subcellular compartmentalization. By engineering a system that allows the ectopic localization of zDHHCs to the outer nuclear membrane (ONM), we show an intriguing substrate specificity of several zDHHCs distinguishing among closely related substrates. Moreover, our data indicate that the steady-state localization of Gαo at the PM and Golgi apparatus is the outcome of local S-palmitoylation events.

These findings contrast the previous view that (i) S-palmitoylation of PMPs occurs exclusively at the Golgi and (ii) serves to drive subsequent PM-directed delivery of such proteins. The unexpected selectivity among different PATs and their substrates we uncover to drive the intracellular localization of PMPs emerges as an attractive target for drug discovery.

## Results

**The minimal localization code in the N-terminus of Gαo.** Early studies using metabolic labeling with [$^3$H]myristate and [$^3$H] palmitate demonstrated that Gαo membrane association is mediated by N-myristoylation at Gly2 and S-palmitoylation at Cys3 in its N-terminus[15–17]. A recent structural analysis identified the recognition sequence of N-myristoyltransferases (NMTs) as an N-terminal hexapeptide, excluding Met1[18]. This suggests that a minimal membrane-binding information might reside within the first seven residues of Gαo. Thus far, three crystal structures of heterotrimeric Go have been solved, showing a prominent α-helix in Gαo N-terminus that extends toward the Gβγ heterodimer. Overlay of the N-termini of these structures revealed the α-helixes to start at position 6 to 10, and to end at position 31 (Supplementary Fig. 1a–c). Similarly, overlay of the N-termini of seven solved structures of Gαi1—a close Gαo homologue—showed the α-helix between the residues 7–8 to 31 (Supplementary Fig. 1d–f). Thus, the N-termini of Gαo and its homologs contain distinct regions: the unstructured lipidated heptapeptide to be followed by the α-helix. To study if these regions have specific roles in Gαo subcellular localization, we generated the following GFP-fusion constructs (Fig. 1a): one including Gαo N-terminal heptapeptide (Gαo-Nt$^7$-GFP), another with the first 31 residues covering the α-helix (Gαo-Nt$^{31}$-GFP), and a third containing only the α-helix (Gαo-Nt$^{8–31}$-GFP). These constructs were expressed in the mouse neuroblastoma Neuro-2a cells (N2a) and their localization at the PM and Golgi apparatus were compared with the full-length Gαo-GFP[5]. Surprisingly, Gαo-Nt$^7$-GFP was predominantly at the Golgi with a weak PM localization, whereas Gαo-Nt$^{31}$-GFP displayed a more homogenous PM and Golgi distribution similar to Gαo-GFP (Fig. 1b–d). Quantification of average fluorescence intensities at these compartments confirmed a much higher Golgi and a lower PM localization of Gαo-Nt$^7$ compared to Gαo-Nt$^{31}$ and Gαo (Fig. 1e, f; see Methods for a detailed description), despite similar expression level of the constructs (Supplementary Fig. 1g, h). Then, we performed a crude subcellular fractionation of N2a cells expressing the constructs and showed that Gαo-Nt$^7$, Gαo-Nt$^{31}$ and Gαo were similarly partitioned between the cytosolic and membrane fractions (Fig. 1g, h). On the other hand, Gαo-Nt$^{8–31}$ was spread over the cytosol and nucleus (Supplementary Fig. 1i), indicating that the N-terminal α-helix alone is not sufficient for membrane association.

In the heterotrimeric G protein complex, the N-terminal α-helix of Gα binds to Gβ[19]. Since Gβ tightly interacts with Gγ, which in turn associates to membranes via its C-terminal prenylation[20], we tested if the poor PM localization of Gαo-Nt$^7$ relates to a lack of Gβγ interaction. For this aim, we co-expressed the Gαo constructs together with mRFP-Gβ1 and mRFP-Gγ3 in N2a cells, and immunoprecipitated the GFP-fusions. Full-length Gαo-GFP strongly pulled down Gβ1γ3, whereas Gαo-Nt$^7$-GFP and Gαo-Nt$^{31}$-GFP showed a very faint co-precipitation of Gβγ with no apparent difference between them (Supplementary Fig. 1j). Thus, the preferential localization of Gαo-Nt$^7$ to the Golgi is independent from Gβγ.

Together, these results indicate that Nt$^7$ is sufficient for Gαo overall membrane binding, but it tends to drive Golgi rather than PM localization.

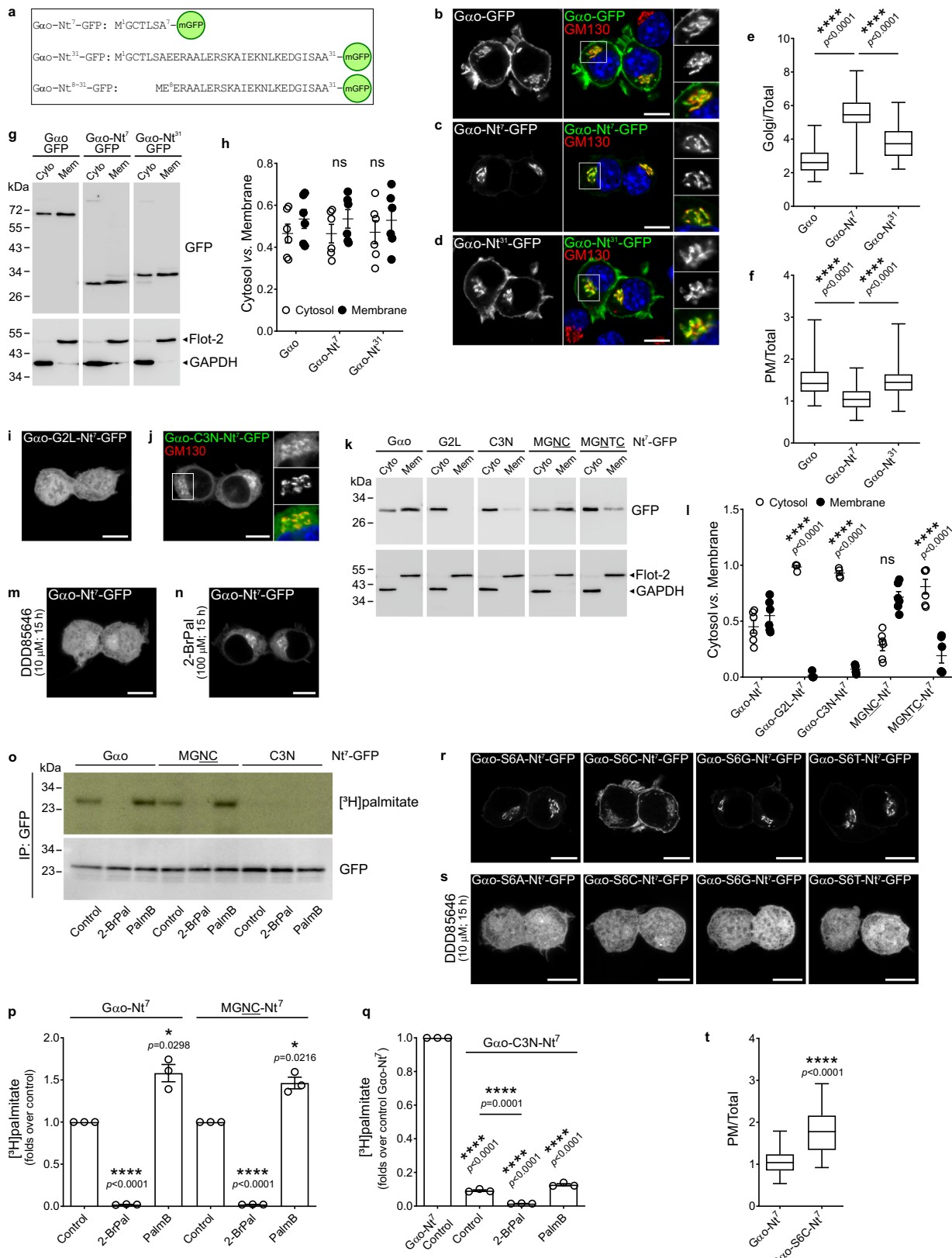

### Key residues in Gαo-Nt[7]

We next aimed to decode the rules of intracellular localization of Gαo-Nt[7] by a systematical point mutation analysis of its residues. Amino acid substitutions were designed using the GPS-Lipid prediction tool[21]. For instance, we introduced the Gly-to-Leu mutation at position 2 of Gαo-Nt[7]-GFP as GPS-Lipid predicted very high likelihood for

S-palmitoylation at Cys3 in the mutant. However, the Gly2 mutation led to a complete loss of membrane association as Gαo-G2L-Nt[7] evenly spread over the cytosol and nucleus (Fig. 1i). This result supports the notion that N-myristoylation is a prerequisite for S-palmitoylation in Gαo[6]. Accordingly, Gαo-G2L-Nt[7] appeared almost exclusively in the cytosolic fraction (Fig. 1k,l).

**Fig. 1 Key amino acid residues in Gαo N-terminus. a** Sequence of the GFP-fusions of Gαo N-terminus. **b–f** N2a cells expressing Gαo-GFP (**b**), Gαo-Nt[7]-GFP (**c**) or Gαo-Nt[31]-GFP (**d**) were immunostained against GM130. Color-channels are listed vertically top-to-bottom and selected areas are magnified to the right with the channels displayed vertically in the same order. DAPI stained nuclei in blue. Mean fluorescence intensity ratios of GFP-fusions at the Golgi (**e**) or PM (**f**) versus total cell. Box plots indicate median (middle line), 25th, 75th percentile (box), and lowest, highest value (whiskers); four independent experiments (Gαo-Nt[7], $n = 56$; Gαo-Nt[31], $n = 61$; Gαo, $n = 58$). **g, h** Subcellular fractionation of constructs described in (**b–d**). Anti-GFP antibody used to detect Gαo constructs, and anti-GAPDH and anti-flotillin-2 (Flot-2) as cytosolic (Cyto) and membrane (Mem) markers, respectively (**g**). Distribution of GFP-fusions in cytosolic and membrane fractions (**h**). Data as mean ± s.e.m.; six independent experiments. **i, j** Localization of Gαo-G2L-Nt[7]-GFP (**i**) and Gαo-C3N-Nt[7]-GFP (**j**) in N2a cells. **k, l** Fractionation of N2a cells expressing Gαo-Nt[7]-GFP or indicated Nt[7]-mutants (**k**). Underlined residues depict substitutions in Gαo-Nt[7]. Immunodetection and quantification done as in (**g, h**). Data represent mean ± s.e.m.; 4–6 independent experiments (**l**). **m, n** Localization of Gαo-Nt[7]-GFP under inhibition of N-myristoylation (DDD85646; **m**) or S-palmitoylation (2-bromopalmitate; 2-BrPal; **n**). **o–q** [³H] palmitate radiolabeling of Gαo-Nt[7]-GFP, MGNC-Nt[7]-GFP, and Gαo-C3N-Nt[7]-GFP. Immunoprecipitated (IP) GFP-fusions from control, 2-BrPal and Palmostatin B (PalmB) treated cells were analyzed by autoradiography ([³H]palmitate) and anti-GFP antibody (**o**). Radioactivity incorporated to constructs normalized to respective controls (**p**) or Gαo-Nt[7] (**q**). Data as mean ± s.e.m.; three independent experiments. **r, s** Localization of Serine 6 (S6) mutants of Gαo-Nt[7]-GFP under control (**r**) or DDD85646 (**s**) treatment. Ser-to-Cys (S6C) mutant showed a higher PM targeting, quantified in (**t**). Box plots indicate median (middle line), 25th, 75th percentile (box), and lowest, highest value (whiskers); four independent experiments (Gαo-Nt[7], $n = 56$; S6C, $n = 63$). **b–d, i, j, m, n, r, s** Scale bars, 10 μm. $P$ values were determined using one-way ANOVA Tukey test for (**e, f**), two-way ANOVA Tukey test for (**h, l**), one-sample $t$-test for (**p, q**), and two-sided unpaired $t$-test for (**q, t**). ns: not significant. Source data are provided as a Source Data file.

We then introduced the Cys-to-Asn mutation at position 3 as GPS-Lipid predicted a high score for N-myristoylation at Gly2 in the mutant. The resulting Gαo-C3N-Nt[7]-GFP was predominantly excluded from the nucleus and showed an ER-like distribution with certain accumulation at Golgi, which might account for the high membrane density of the cisternae stack (Fig. 1j). Nevertheless, Gαo-C3N-Nt[7] appeared mostly in the cytosolic fraction (Fig. 1k, l), confirming that N-myristoylation confers only weak membrane-binding properties[7]. The distinct localization patterns of Gαo-Nt[7] mutants impaired in N-myristoylation and S-palmitoylation were not associated with significant variations in their expression levels (Supplementary Fig. 1k, l), and were emulated by the treatment with the specific inhibitors DDD85646[22] and 2-bromopalmitate (2-BrPal)[23], respectively (Fig. 1m, n). Defects in S-palmitoylation of Gαo-G2L-Nt[7] and Gαo-C3N-Nt[7] were confirmed by metabolic labeling with [³H] palmitate (Supplementary Fig. 1m): only a low level of [³H]palmitate incorporation remained for both mutants. These remaining signals disappeared upon cleavage of fatty acyl thioester bonds by hydroxylamine (Supplementary Fig. 1m), implying that the G2L and C3N GFP fusions were indeed S-palmitoylated. However, as the only Cys residues present in Gαo-C3N-Nt[7] lay within the GFP sequence (Cys48 and Cys70), these results indicate that the remaining level of S-palmitoylation observed in these Nt[7] constructs originated from GFP. Quantification of [³H]palmitate labeling confirmed the drastic reduction in S-palmitoylation of Gαo-Nt[7] upon 2-BrPal treatment, and the weak labeling of GFP in the C3N mutant (~10% of Gαo-Nt[7]); a signal further reduced by 2-BrPal incubation (Fig. 1o–q). We similarly performed [³H]palmitate metabolic labeling of the full-length Gαo-GFP and its C3N mutant. As expected, 2-BrPal changes the localization of full-length Gαo and almost abolished its [³H]palmitate signal (Supplementary Fig. 1n–p), yet a weak labeling was still observed for the Gαo-C3N mutant (~10% of Gαo) that was additionally reduced by 2-BrPal (Supplementary Fig. 1o–q). This marginal S-palmitoylation of GFP in the G2L and C3N constructs is likely stochastic and does not contribute to their membrane association (Fig. 1k, l), thus bearing no influence on localization of the Gαo constructs in our study.

The most recent substrate recognition motif for NMTs was described as M[1]G[^DEFRWY]X[^DEKR][ACGST][KR][7], with ^ denoting exclusion of residues, and X—any amino acid[18]. The authors did not exclude the possibility that peptides lacking a positive charge at position 7 might also be accepted by NMTs. Thus, N-myristoylation of Gαo—and all human N-myristoylated Gα subunits—is not in conflict with the lack of a Lys/Arg residue at

position 7 (Supplementary Fig. 1a, d). At position 6, a Ser residue is found in Gαo and its homologues, and Ser is by far the most frequent residue at this position among NMT substrates[18]. Thus, we first introduced point mutations at Ser6 in Gαo-Nt[7] based on the substrate recognition motif described by Castrec et al. and on a previous study indicating that Arg, Asn, Phe, and Val can occupy this position as well[24]. Substitution of Ser6 by Ala, Cys, Gly, or Thr showed efficient membrane binding (Fig. 1r) and expression levels similar to Gαo-Nt[7] (Supplementary Fig. 2a, b). Interestingly, the S6C mutant displayed an increased PM localization as compared to the wild-type Gαo-Nt[7]-GFP (Fig. 1r, t) and a migration shift in SDS-polyacrylamide gels (Supplementary Fig. 2a), suggesting that Cys3 and Cys6 both undergo S-palmitoylation and that PM targeting might be enhanced by the dual S-palmitoylation. N-myristoylation of these mutants was indirectly verified with the inhibitor DDD85646 (Fig. 1s). Conversely, we observed a largely cytosolic and nuclear localization for the constructs in which the Ser6 was substituted by Arg, Asn, Phe and Val (Supplementary Fig. 2c), suggesting that such amino acid substitutions make the N-terminal peptide of Gαo a poor substrate for MNTs; the expression levels of the mutant constructs did not significantly differ from Gαo-Nt[7] (Supplementary Fig. 2a, b). We conclude that the lipid modifications and subcellular localization of the Gαo-Nt[7] constructs are highly sensitive to even minor changes in this heptapeptide sequence.

**Cysteine position**. Intrigued by the S6C mutant, we questioned if the position of the Cys residue within Gαo-Nt[7] might play any role in its subcellular localization. Thus, we moved the Cys3 to positions 4 (MGNC-Nt[7]) and 5 (MGNTC-Nt[7]), and expressed the GFP constructs in N2a cells. An Asp residue was placed at position 3 following GPS-Lipid prediction. Unexpectedly, MGNC-Nt[7]-GFP displayed a completely different localization than its parental Gαo-Nt[7] (Figs. 1c and 2a), showing PM localization that was not only higher than that of Gαo-Nt[7]-GFP, but also of the full-length Gαo-GFP (Fig. 2c). Moreover, the perinuclear structures labeled by MGNC-Nt[7] showed a much lower co-localization with the Golgi marker GM130 compared to Gαo-Nt[7] (Pearson's correlation; Figs. 1c and 2a, d). Instead, the perinuclear structures containing MGNC-Nt[7] co-localized better with Lactadherin-C2 (mRFP-Lact-C2; Supplementary Fig. 2d, e), a biosensor for phosphatidylserine[25]. These results imply that MGNC-Nt[7] preferentially associates with the PM-derived endocytic rather than the Golgi-derived secretory pathway. The distinct localization of MGNC-Nt[7] was not the result of changed expression or membrane binding, as its level (Supplementary Fig. 1k, l) and presence in the membrane fraction

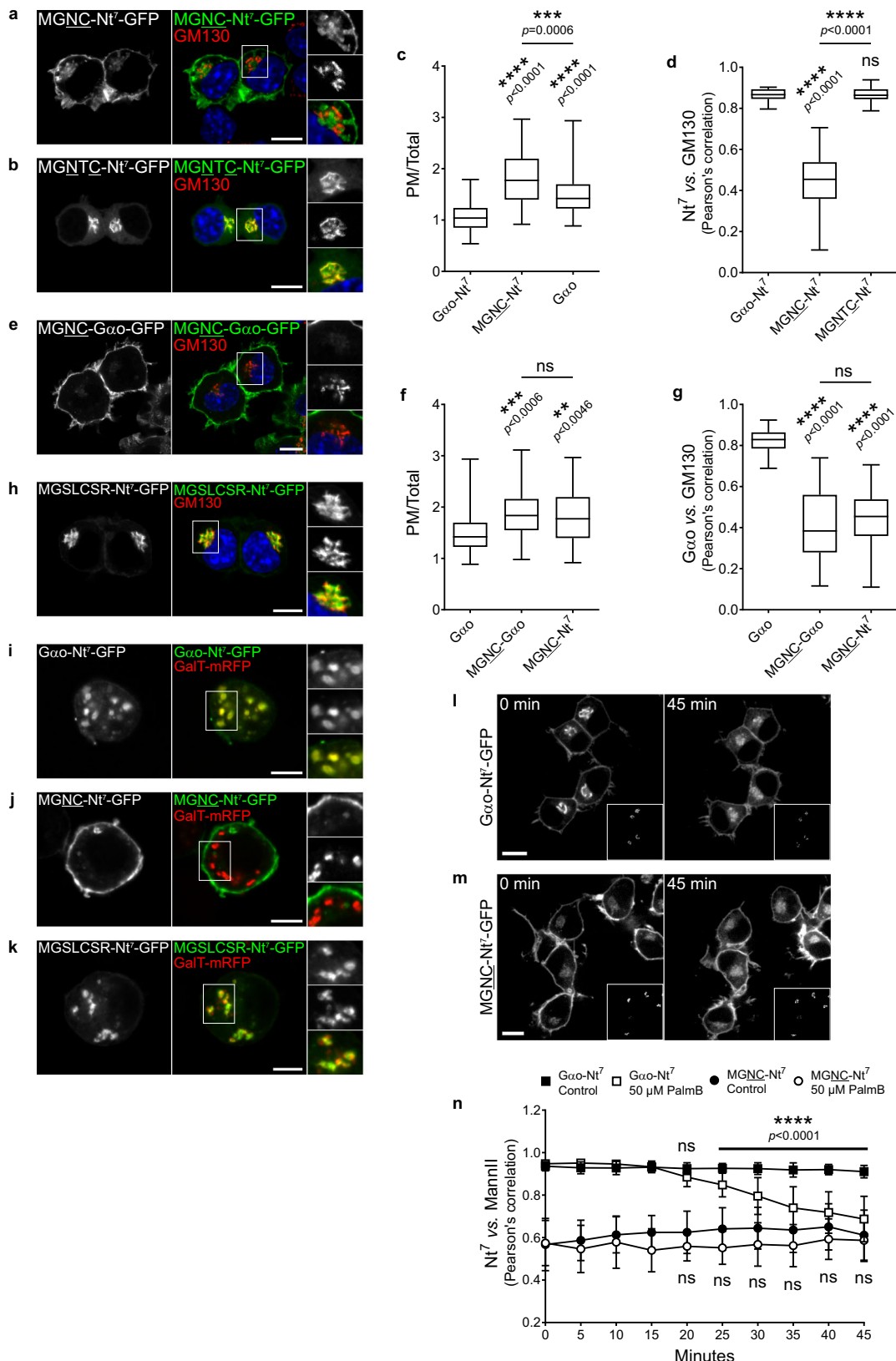

(Fig. 1k, l) were not significantly different from that of Gαo-Nt[7]. On the other hand, the Cys5 construct MGNTC-Nt[7]-GFP localized at the Golgi and cytosol (Fig. 2b, d) but appeared mostly in the cytosolic fraction upon cell fractionation (Fig. 1k, l). Thus, it appears that Cys5 is not a good substrate for PATs, as MGNTC-Nt[7] closely resembles the C3N mutant (Fig. 1j–l).

Accordingly, metabolic labeling showed a wild-type-like [3H] palmitate incorporation for MGNC-Nt[7] and only the residual signal for MGNTC-Nt[7] (Supplementary Fig. 1m). S-palmitoylation of MGNC-Nt[7] was additionally confirmed by its sensitivity to the inhibitor 2-BrPal (Fig. 1o, p and Supplementary Fig. 2f).

**Fig. 2 Position of the Cys in Gαo N-terminus. a–d** N2a cells expressing the Nt[7]-GFP mutants MGNC (**a**) or MGNTC (**b**) and immunostained against GM130 (**a**, **b**). Nuclei in blue stained with DAPI. Color-channels are listed vertically top-to-bottom and selected areas are magnified to the right with the channels displayed vertically in the same order. Underlined letters indicate residues substituted in Gαo-Nt[7]. Mean fluorescence intensity ratios of GFP-fusions at the PM versus total cell (**c**); four independent experiments (Gαo-Nt[7], $n = 56$; MGNC-Nt[7], $n = 50$; Gαo, $n = 58$). Co-localization analysis of the Nt[7]-GFP constructs with GM130 (**d**); three independent experiments (Gαo-Nt[7], $n = 33$; MGNC-Nt[7], $n = 36$; MGNTC-Nt[7], $n = 39$). **e–g** N2a cells expressing the MGNC mutant of full-length Gαo (MGNC-Gαo-GFP) and immunostained against GM130 (**e**). Boxed area is enlarged. Underlined letters indicate residues substituted in Gαo. Relative localization of GFP-fusions at the PM (**f**); four independent experiments (Gαo, $n = 56$; MGNC-Gαo, $n = 46$; MGNC-Nt[7], $n = 50$). Co-localization with GM130 (**g**); three independent experiments (Gαo, $n = 47$; MGNC-Gαo, $n = 38$; MGNTC-Nt[7], $n = 36$). **h** Localization in N2a cells of a Nt[7]-GFP construct comprising the consensus sequence of eukaryotic Gα-Nt[7] with the conserved Cys at position 5 (MGSLCSR). Marked region is magnified to the right. **i–k** *Drosophila* S2 cells expressing Gαo-Nt[7]-GFP (**i**), MGNC-Nt[7]-GFP (**j**) or MGSLCSR-Nt[7]-GFP (**k**). Co-expression of GalT-mRFP marked Golgi stacks (**i–k**). Selected areas are zoomed-in to the right. **l–n** Live imaging of N2a cells co-expressing Gαo-Nt[7]-GFP (**l**) or MGNC-Nt[7]-GFP (**m**) with the Golgi marker MannII-mRFP (bottom right insets). Representative images at the time of Palmostatin B (PalmB) addition (0 min) and after 45 min. Co-localization of Nt[7]-GFP constructs with MannII-mRFP in 5 min intervals and starting at $t = 0$ (**n**). Data represent mean ± s.d. (Gαo-Nt[7] control, $n = 20$; Gαo-Nt[7] PalmB, $n = 20$; MGNC-Nt[7] control, $n = 16$; MGNC-Nt[7] PalmB, $n = 20$). **a**, **b**, **e**, **h**, **l**, **m** Scale bars, 10 μm; **i–k** Scale bars, 5 μm. **c**, **d**, **f**, **g** Box plots indicate median (middle line), 25th, 75th percentile (box), and lowest, highest value (whiskers). *P* values were determined using one-way ANOVA Tukey test for (**c**, **d**, **f**, **g**) and one-way ANOVA Šídák test for **n**. ns: not significant. Source data are provided as a Source Data file.

Intriguingly, the subcellular localization of MGNC-Nt[7] was fully recapitulated in the full-length Gαo when the same mutation was introduced. MGNC-Gαo-GFP showed a higher PM targeting (Figs. 1b and 2e, f) and a much lower co-localization with the Golgi marker GM130 (Figs. 1b and 2e, g) than Gαo-GFP despite comparable levels of expression (Supplementary Fig. 2g, h). The S-palmitoylation of MGNC-Gαo was confirmed by metabolic labeling with [3H]palmitate (Supplementary Fig. 1o, p). As expected, the localization and [3H]palmitate labeling of MGNC-Gαo were strongly affected by the 2-BrPal treatment (Supplementary Figs. 1o, p and 2i).

Altogether, these data uncover a remarkable characteristic of the N-terminal localization code of Gαo: minor changes in its amino acid sequence can drastically modify the subcellular localization of the protein. Due to its homology with all dual-lipidated Gα subunits, and to other similarly modified PMPs, we reasoned that this principle might have a broad and evolutionary conserved relevance in biology.

**The evolution of Gα-Nt[7].** We wanted to determine if Cys residues might exist at different positions within Nt[7] in eukaryotic Gα subunits. We searched for Gα sequences containing the N-myristoylation signature M[1]GXXX[ACGST]X[7] and found Cys throughout positions 3 to 5 (Supplementary Methods, Supplementary Notes, Supplementary Fig. 3 and Supplementary Data 1). Interestingly, a consensus sequence—MGSLCSR—emerged only for Gα sequences containing Cys at position 5 (Supplementary Notes, Supplementary Fig. 3 and Supplementary Data 2). Thus, we expressed a GFP-fusion of this sequence in N2a cells. Resultingly, MGSLCSR-Nt[7]-GFP showed stronger membrane binging than the initially designed MGNTC-Nt[7] localizing predominantly at the Golgi with low cytosolic signal (Fig. 2h) despite similar expression levels (Supplementary Fig. 1k, l). When MGSLCSR was introduced into the full-length Gαo, however, the cytosolic localization of this mutant was well visible (Supplementary Fig. 2j), although its expression was comparable to Gα-GFP (Supplementary Fig. 2g, h). This might probably relate to the fact that Gα subunits carrying a Cys at position 5 are virtually absent in Metazoa (Supplementary Notes and Supplementary Data 1).

Then, we expressed our constructs Gαo-Nt[7]-GFP, MGNC-Nt[7]-GFP and the consensus MGSLCSR-Nt[7]-GFP in the *Drosophila* Schneider-2 cell line. Remarkably, all constructs retained the same localization patterns as seen in N2a cells: Gαo-Nt[7] and MGSLCSR-Nt[7] targeted mainly Golgi stacks labeled by GalT-mRFP, while MGNC-Nt[7] associated largely with the PM

(Fig. 2i–k). Similar localizations were also seen in HeLa cells (Supplementary Fig. 2k). These data point to highly conserved rules for substrate compartmentalization across cell types and species.

**Direct targeting of Nt[7] variants to Golgi or PM.** Our next goal was to understand the molecular mechanism behind the contrasting subcellular localization of two rather similar sequences: Gαo-Nt[7] and MGNC-Nt[7] (Figs. 1c and 2a). N-myristoylation generally occurs during protein synthesis and confers a weak membrane binding to the modified protein[23], and is essential for the subsequent S-palmitoylation to occur[6]. Our own data showed that myristoylated, but not palmitoylated Gαo-Nt[7] and MGNC-Nt[7] are indistinguishable in their localization and only prevented from free cytosolic and nuclear diffusion (Fig. 1n and Supplementary Fig. 2f). Therefore, we speculated that S-palmitoylation is key for subcellular compartmentalization, by itself or in combination with the myristoyl group and/or neighboring amino acids. It has been suggested that S-palmitoylation of PMPs—including Gα subunits—occurs exclusively at the Golgi, an event followed by their transport to the PM via the secretory pathway[12]. In addition to the Golgi-to-PM trafficking, a proper steady-state localization of PMPs is controlled by their rapid and ubiquitous depalmitoylation[12]. Thus, we next explored whether the distinct localizations of Gαo-Nt[7] and MGNC-Nt[7] might involve a differential depalmitoylation and/or Golgi-to-PM transport of the constructs.

We performed live imaging of N2a cells expressing Gαo-Nt[7]-GFP or MGNC-Nt[7]-GFP during incubation with the depalmitoylation blocker Palmostatin B, an acyl protein thioesterase 1 and 2 inhibitor[26]. Localization of Gαo-Nt[7] and MGNC-Nt[7] in control cells was not affected during the 45 min of recording (Supplementary Fig. 2l, m), whereas both constructs showed progressive changes upon Palmostatin B addition. Particularly, the signal of Gαo-Nt[7] at Golgi diffused over the surrounding area upon time, whereas its PM localization seemed not to be affected (Fig. 2l and Supplementary Movie 1). Accordingly, quantification revealed that its PM content did not significantly change during the entire recording time (Supplementary Fig. 2n), but its co-localization with the MannII-mRFP Golgi marker showed a significant decrease starting at 25 min of treatment (Fig. 2n). The PM signal of MGNC-Nt[7] presented no major changes during Palmostatin B treatment as well, but its overall presence in the perinuclear region slightly increased in a pattern that did not resemble the Golgi (Fig. 2m and Supplementary Movie 2). In fact, MGNC-Nt[7] co-localization with the Golgi marker and PM

content did not significantly change during the treatment (Fig. 2n and Supplementary Fig. 2o). Comparable changes in localization were observed for the full-length Gαo-GFP and MGNC-Gαo-GFP after 45 min incubation with Palmostatin B (Supplementary Fig. 2p, q). Metabolic labeling showed the efficacy of Palmostatin B, as [³H]palmitate incorporation appears similarly augmented for all constructs (Fig. 1o, p and Supplementary Fig. 1o, p). This analysis supports the notion that the steady-state localization of PMPs results from a palmitoylation/depalmitoylation equilibrium[12]. However, the fact that the PM content of Gαo-Nt⁷ was not affected by Palmostatin B argues against a constant Golgi-to-PM flow of the construct.

We next aimed at visualizing how Nt⁷ constructs are trafficked in the cell. We adapted to our constructs the reverse dimerization (RD) assay, originally designed for the synchronized trafficking of secretory and integral membrane proteins[27]. This method is based on the aggregation of the F36M mutant of FKBP12 (FM) that allows protein tracking upon chemical dissociation. Thus, we intercalated four FM copies (FM⁴) between Nt⁷ and GFP and performed the RD assay in HeLa cells as their larger cell bodies grant for a better visualization. The distinct localization of Gαo-Nt⁷ and MGNC-Nt⁷—still observed in HeLa cells (Supplementary Fig. 2k)—was lost by the FM⁴ insertion in both cell lines, and cytosolic aggregates of different sizes were visualized instead. After chemical dissociation by the D/D solubilizer, most of the aggregates vanished and the constructs showed their characteristic localizations (Fig. 3a, b).

Then, live imaging was performed in HeLa cells co-expressing either Nt⁷-FM⁴-GFP construct together with MannII-mRFP as Golgi marker. As expected, Gαo-Nt⁷-FM⁴-GFP showed a rapid accumulation at the Golgi once the D/D solubilizer was added (Fig. 3c, e and Supplementary Movie 3). On the other hand, MGNC-Nt⁷-FM⁴-GFP presented a rather slow and poor accumulation at the Golgi, as opposed to the rapid PM targeting of the construct (Fig. 3d, e and Supplementary Movie 4). This result does not support a Golgi-to-PM flow of MGNC-Nt⁷-FM⁴-GFP, as PM targeting was not preceded by its Golgi accumulation. To further prove that PM localization of MGNC-Nt⁷-FM⁴-GFP is indeed independent from transport through Golgi, we performed the RD assay under the 20 °C temperature block that forces cargos to accumulate in Golgi[28]. As control, we used the GFP-FM⁴-hGH construct that aggregates in the ER lumen and is secreted via Golgi-mediated transport after chemical dissociation[29]. Incubation with D/D solubilizer for 45 min induced an almost complete secretion of GFP-FM⁴-hGH at 37 °C, while at 20 °C the construct strongly accumulated in the Golgi (Fig. 3f). Conversely, HeLa cells expressing MGNC-Nt⁷-FM⁴-GFP showed a comparable PM localization of the construct at both temperatures (Fig. 3g). Quantification of MGNC-Nt⁷-FM⁴-GFP mean fluorescence intensity in the MannII-mRFP region revealed no significant difference in its Golgi content at 37 and 20 °C (Fig. 3h), indicating that PM targeting of MGNC-Nt⁷ does not occur via transport through the Golgi.

Altogether, these results imply that the characteristic steady-state localizations of Gαo-Nt⁷ and MGNC-Nt⁷ are not related to Golgi-to-PM trafficking. Instead, each of the constructs goes directly to its primary destination: Golgi or PM.

**Lipid binding of Nt⁷ variants does not explain the preferential localization**. Thus far, our data indicate that S-palmitoylation might account for the compartmentalization of Gαo-Nt⁷ and MGNC-Nt⁷, and that intracellular trafficking is not a major player in their distinctive localizations. We then envisioned two further scenarios that could explain the differential Nt⁷ compartmentalization: (i) PATs have no specificity for substrate

recognition, and substrates concentrate at different compartments due to specific interactions, and (ii) PATs discriminate substrates, which in turn accumulate at the compartment where S-palmitoylation takes place. The first scenario involves a certain degree of promiscuity among PATs[11], and implies that lipidations and a few surrounding amino acids confer specific binding properties, e.g., to lipids, which are known to differentially accumulate all over the endomembrane system[30]. In addition, fatty acids other than palmitate can also be attached to the available Cys and might confer different binding specificities to the S-acylated protein[31]. The second scenario requires that PATs discriminate between analogous substrates as small as Nt⁷ and that a given substrate accumulates at the compartment where its specific PAT is localized.

To test the first scenario, we used membrane strips spotted with fifteen different lipids found in cellular membranes and performed a protein-lipid overlay assay[32]. As control, we employed a GFP-fusion of the pleckstrin homology (PH) domain of FAPP1 (FAPP1-PH-GFP), which specifically binds to phosphatidylinositol 4-phosphate (PI4P)[33]. N2a cells were transfected with Gαo-Nt⁷-GFP, MGNC-Nt⁷-GFP or FAPP1-PH-GFP, and then membrane strips were incubated with cleared cell extracts. As expected, FAPP1-PH showed a strong binding to the spot containing PI4P (Supplementary Fig. 4a). On the other hand, no apparent difference was found for Gαo-Nt⁷ and MGNC-Nt⁷ that mainly bound to phosphatidylserine, cardiolipin, and phosphoinositides, particularly to PI4,5P2 (Supplementary Fig. 4a). A similar lipid-binding pattern was also observed by the larger construct Gαo-Nt³¹-GFP (Supplementary Fig. 4a). Thus, the compartmentalization of Gαo-Nt⁷ and MGNC-Nt⁷ is unlikely to emerge from different lipid-binding affinities, arguing against the first scenario.

**A tool for visualization of local S-palmitoylation in intact cells**. Next, we studied the second scenario, which implies that local S-palmitoylation drives substrate compartmentalization. In mammals, intracellular S-palmitoylation is mediated by zDHHC proteins with PAT activity; 23 zDHHCs exist in humans and 24 in mice[10]. Most of the zDHHCs have four transmembrane domains (TMDs), whereas only two and three members present five and six TMDs, respectively[11]. The majority of zDHHCs localize at the Golgi and the remaining are distributed among the ER, PM, and endosomes[34,35]. In order to confirm their subcellular localization, we expressed a collection of twenty-three HA-tagged mouse zDHHCs in N2a cells[36]. As expected, immunostainings revealed that a large number of zDHHCs localized predominantly at the Golgi: zDHHC3, 7, 9, 12, 13, 15–17, 21, 23, and 25 (Supplementary Fig. 4b). Six showed a clear PM localization but were also found on endosomes: zDHHC2, 5, 11, 14, 18, and to a lesser extent zDHHC8. The rest of the PATs mostly showed ER localization, while zDHHC4 was associated almost exclusively with the nuclear envelope (Supplementary Fig. 4b), a pattern not described in previous studies[34,35].

Besides a unique antibody that specifically recognizes palmitoylated PSD-95[37], experimental approaches to study zDHHC-substrate pair relations are based exclusively on disruptive biochemical and proteomic techniques[38]. To determine if Gαo-Nt⁷ and MGNC-Nt⁷ are differently lipidated at the PM vs. Golgi, we aimed at developing a method to visualize local S-palmitoylation in intact cells. Inspired by the nuclear envelope localization of zDHHC4, we engineered zDHHCs that ectopically target the ONM to detect mislocalized substrates. We took advantage of some components of the well-studied LINC (linker of nucleoskeleton and cytoskeleton) complexes. LINC complexes are built by proteins of the KASH (Klarsicht, ANC-1, and Syne

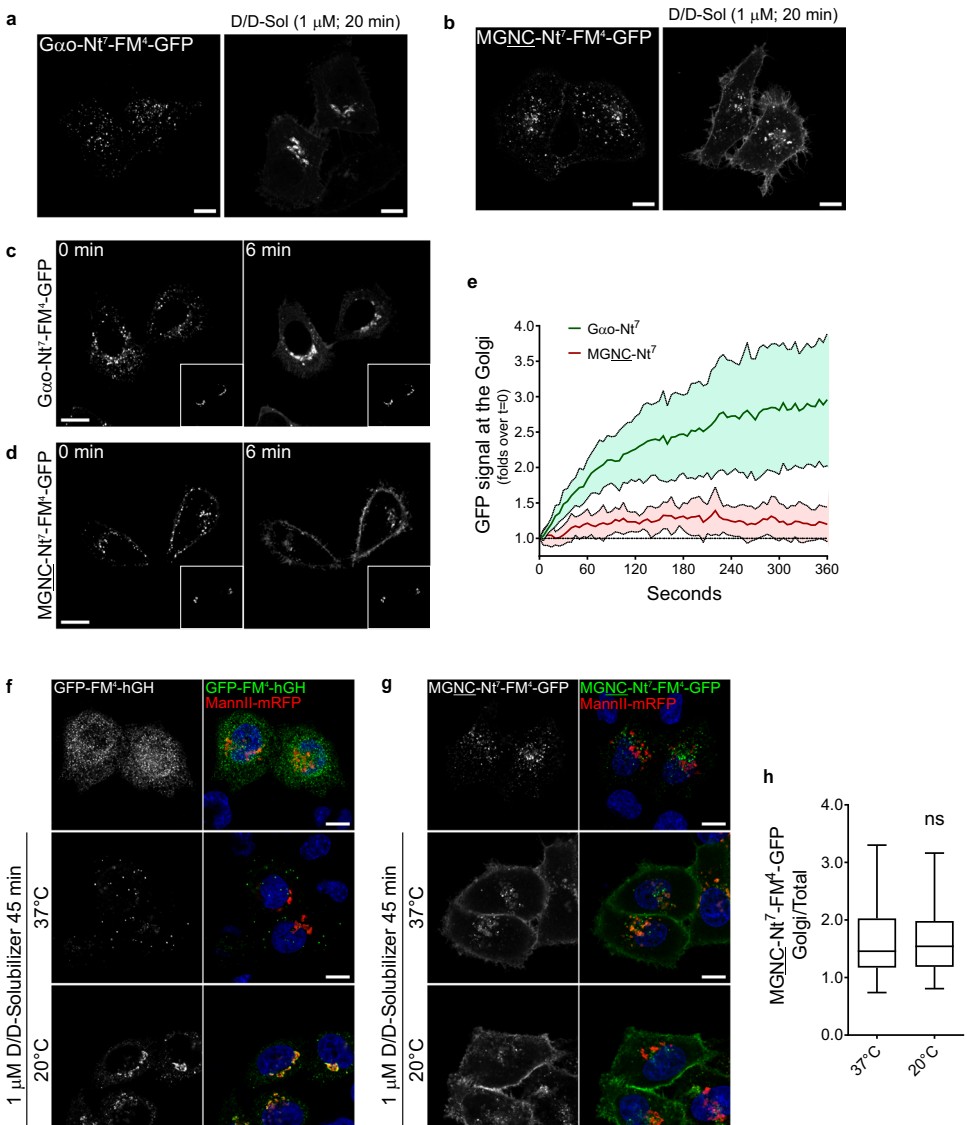

**Fig. 3 Targeting of Nt⁷-GFP. a, b** Reverse dimerization (RD) assay in HeLa expressing Gαo-Nt$^7$-FM$^4$-GFP (**a**) or MGNC-Nt$^7$-FM$^4$-GFP (**b**). Cytosolic clusters formed by the Nt$^7$-FM$^4$-GFP constructs dissolve by the addition of D/D solubilizer (D/D-Sol) and show their expected localizations. Underlined letters indicate residues substituted in Gαo-Nt$^7$. **c–e** Live imaging of HeLa cells co-expressing Gαo-Nt$^7$-FM$^4$-GFP (**c**) or MGNC-Nt$^7$-FM$^4$-GFP (**d**) and the Golgi marker MannII-mRFP (bottom right insets). Representative images of the Nt$^7$-FM$^4$-GFP constructs at the time of D/D solubilizer addition (0 min) and after 6 min (**c, d**). Increase upon time of Nt$^7$-FM$^4$-GFP constructs at the Golgi region (**e**). Note that Gαo-Nt$^7$-FM$^4$-GFP quickly targets the Golgi region, while MGNC-Nt$^7$-FM$^4$-GFP only slightly accumulates at the Golgi. Data represent mean ± s.d. (Gαo-Nt$^7$, $n = 12$; MGNC-Nt$^7$, $n = 11$). **f–h** RD assay in HeLa cells expressing the secretable control GFP-FM$^4$-hGH (**f**) or MGNC-Nt$^7$-FM$^4$-GFP (**g**) performed at 37 °C or at 20 °C to inhibit Golgi transport. Incubation with D/D solubilizer results in the almost complete secretion of GFP-FM$^4$-hGH at 37 °C but in Golgi retention at 20 °C (**f**). PM targeting of MGNC-Nt$^7$-FM$^4$-GFP seems not affected by the 20 °C temperature block and its presence at the Golgi region (MannII-mRFP) is not higher than at 37 °C (**g**). Mean fluorescence intensity of MGNC-Nt$^7$-FM$^4$-GFP, Golgi over total cell (**h**). Box plots indicate median (middle line), 25th, 75th percentile (box), and lowest, highest value (whiskers); three independent experiments (37 °C, $n = 49$; 20 °C, $n = 40$). **a–d, f, g** Scale bars, 10 μm. $P$ value was determined using a two-sided unpaired $t$-test for (**h**). ns: not significant. Source data are provided as a Source Data file.

Homology) family that are embedded in the ONM and interact within the perinuclear space with proteins of the SUN (Sad1 and UNC-84) family, which in turn are inserted in the inner nuclear membrane[39]. As the C-termini of all zDHHCs except zDHHC24 face the cytosol, we hypothesized that the addition of a C-terminal KASH-domain might induce their targeting to the ONM (Fig. 4a). To test this hypothesis, we first generated a zDHHC5 construct carrying an N-terminal mRFP for visualization and the KASH-domain of syne-1/nesprin-1 at its C-terminus (mRFP-zDHHC5-KASH). In HeLa cells, the KASH-domain alone (mRFP-KASH) is efficiently targeted to the ONM (Fig. 4b),

and mRFP-zDHHC5 localized at the PM and endosomes as expected (Fig. 4c). The addition of the KASH domain strongly impaired PM localization of zDHHC5 (Fig. 4d), but a robust ONM targeting was only achieved by the co-expression of the KASH-interacting protein SUN2 (Fig. 4e).

Next, we analyzed if known zDHHC-substrate pair relations can be recapitulated in HeLa cells and chose the PMPs SNAP23, caveolin-1 and flotillin-2. The core of SNAP23 contains 6 Cys residues that are palmitoylated by zDHHC13 and 17[40]. zDHHC7 appears as the main PAT for caveolin-1, which is palmitoylated in 3 Cys residues at its C-terminus[41]. Flotillin-2 is myristoylated at

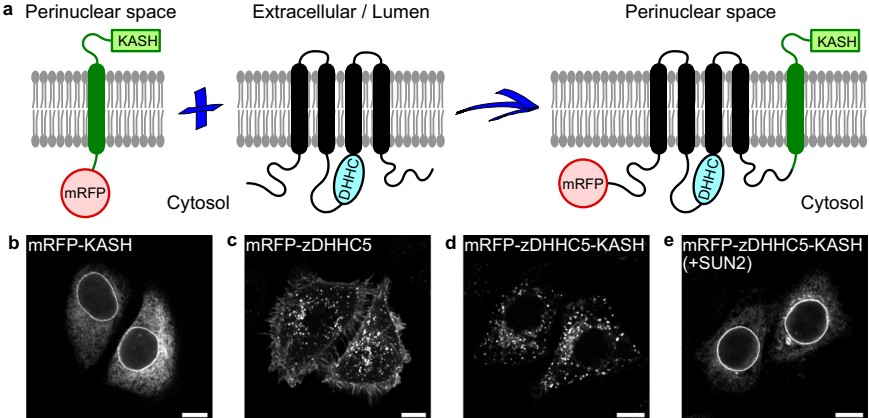

**Fig. 4 Local S-palmitoylation at the ONM—the "SwissKASH" assay. a** Schematic representation of the core components of the SwissKASH assay. **b**–**e** Representative images of HeLa cells expressing the constructs mRFP-KASH that targets the ONM (**b**), mRFP-zDHHC5 (**c**) or mRFP-zDHHC5-KASH (**d**, **e**) with or without the co-expression of SUN2. Note that mRFP-zDHHC5-KASH efficiently targets the ONM in the presence of SUN2 (**e**). **b**–**e** Scale bars, 10 μm.

Gly2 and palmitoylated at Cys4, 19, and 20 by zDHHC5[42]. Remarkably, the GFP-fusions of SNAP23 (Supplementary Fig. 5a), caveolin-1 (Supplementary Fig. 5b) and flotillin-2 (Supplementary Fig. 5c) were efficiently recruited to the ONM by the corresponding zDHHC-KASH constructs. Together, we successfully developed a system in intact cells for the visualization of local S-palmitoylation at the ONM and named it the "SwissKASH" assay.

**S-palmitoylation of Nt[7] at the ONM—the SwissKASH assay**. Once the SwissKASH assay was established, we first tested if PM-associated zDHHCs showed differential substrate specificities for Gαo-Nt[7] and MGNC-Nt[7]. In addition to zDHHC5, the C-terminal KASH-domain effectively targeted zDHHC2, 8, 11, 14, and 18 to the ONM (Fig. 5a–d and Supplementary Fig. 5d, e). Remarkably, Gαo-Nt[7]-GFP and MGNC-Nt[7]-GFP were efficiently recruited to the ONM by co-expression of mRFP-zDHHC11-KASH but not by the control mRFP-KASH (Fig. 5a–d). To quantify these effects, we measured the mean GFP-fluorescence intensity at the ONM and nearby cytosol, and used their ratio to determine the relative ONM content of the Nt[7] constructs. Quantification revealed a similar ~2.5-fold accumulation of Gαo-Nt[7] and MGNC-Nt[7] at the ONM by co-expression of mRFP-zDHHC11-KASH (Fig. 5e), suggesting that zDHHC11 equally accepts both Nt[7] constructs as substrate. On the other hand, the KASH-fusions of zDHHC2, 5, 8, 14, and 18 showed no effect on the localization of Gαo-Nt[7] and MGNC-Nt[7] (Supplementary Fig. 5d,e). Thus, it appears that the predominant PM localization of MGNC-Nt[7] cannot be explained by major differences in substrate specificity of PM-associated zDHHCs. Our data also identifies zDHHC11 as the main PM-localized PAT for Gαo.

Next, we applied the SwissKASH system to Gαo-Nt[7] and MGNC-Nt[7] using the eleven Golgi-associated zDHHCs, as all were efficiently targeted to the ONM by the C-terminal KASH-domain (Fig. 5f–i and Supplementary Fig. 6a, b). Notably, mRFP-zDHHC-KASH constructs for zDHHC3 and 7 caused strong ONM accumulation of Gαo-Nt[7]-GFP (Fig. 5f, g). No other zDHHCs showed any activity toward Gαo-Nt[7] (Supplementary Fig. 6a), suggesting a high degree of specificity in substrate recognition among Golgi-associated PATs and identifying zDHHC3 and 7 as the main Golgi-localized PATs for Gαo. MGNC-Nt[7]-GFP, in contrast, was recruited to the ONM only to a much lower extent by the KASH-fusions of zDHHC3 and 7 (Fig. 5h, i); the remaining Golgi zDHHC-KASH constructs were

inefficient in ONM recruitment (Supplementary Fig. 6b). Quantification showed that Gαo-Nt[7] accumulates at the ONM roughly 2.0 and 2.5 folds by the co-expression of zDHHC3 and zDHHC7, respectively (Fig. 5j, k). MGNC-Nt[7] presented a significantly lower ONM recruitment of ~1.5 folds by zDHHC3 and 7 (Fig. 5j, k). To confirm that the PAT activity was responsible for the mislocalization of substrates in the SwissKASH assay, we generated zDHHC-inactive constructs by a Cys-to-Ser mutation in the catalytic DHHC domain[36]. As anticipated, the ability of the KASH-fusions of zDHHC3, 7, and 11 to induce ONM accumulation of Gαo-Nt[7] and MGNC-Nt[7] was completely abolished when DHHS-mutants were co-expressed (Fig. 5l–o).

As the KASH-fusion of zDHHC11 was able to target both Gαo-Nt[7] and MGNC-Nt[7] to the ONM (Fig. 5b, d, e), we asked whether it acts in an indiscriminative manner or can still distinguish among different Nt[7] substrates. Answering this question, we found that mRFP-zDHHC11-KASH induced a strong ONM recruitment of MGSSCSR-Nt[7]-GFP but showed a much weaker effect on MGLLCSR-Nt[7]-GFP (Supplementary Fig. 7a–c), both sequences commonly found in plant Gα subunits (Supplementary Notes and Supplementary Data 1).

The data above speak for the unexpected specificity of different zDHHCs to Nt[7] substrates that results in specific substrate localizations. We wondered whether these effects could simply result from physical interactions, differential among different zDHHC-substrate pairs. However, we identified that the zDHHCs do not possess high-affinity binding abilities to Nt[7] sequences, as immunoprecipitation of Nt[7]-GFP constructs did not co-precipitate any of the relevant mRFP-zDHHC-KASH constructs (Supplementary Fig. 7d–g).

Thus, we conclude that zDHHC3, 7 and 11 all can differentiate among Nt[7] substrates with even slight variations in amino acid composition.

**S-palmitoylation of Nt[7] at the ONM—the role of accessory proteins**. In recent years, a few accessory proteins have been implicated in the regulation of several zDHHCs[43]. Here, we focused on two closely related PMPs: GCP16/Golga7 which has been reported to interact with the PM-localized zDHHC5[44] as well as the Golgi-localized zDHHC9[45], and Golga7b which associates with zDHHC5 only[46]. We added a Flag-tagged GCP16/Golga7 construct to the SwissKASH system for zDHHC5 and 9, and applied it to Gαo-Nt[7] and MGNC-Nt[7]. Interestingly, GCP16/Golga7 was strongly recruited to the ONM by the KASH-fusion

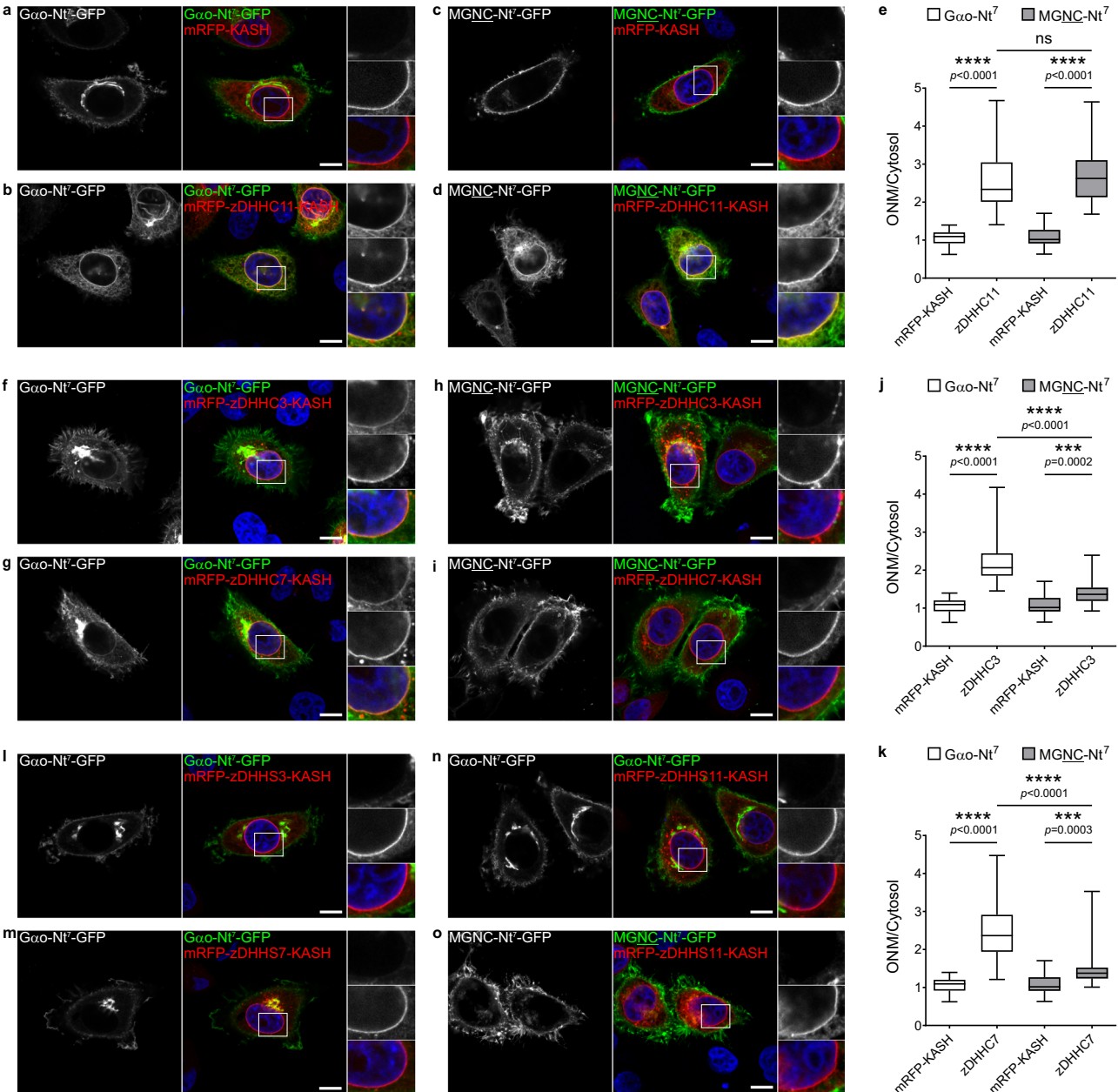

**Fig. 5 The SwissKASH assay for PM and Golgi zDHHCs. a–e** Representative images of the SwissKASH assay in HeLa cells for Gαo-Nt[7]-GFP (**a**, **b**) and MGNC-Nt[7]-GFP (**c**, **d**) using the control mRFP-KASH construct (**a**, **c**) or mRFP-zDHHC11-KASH (**b**, **d**). Nuclei stained in blue with DAPI. Color-channels are listed vertically top-to-bottom and selected areas are magnified to the right with the channels displayed also vertically in the same order. Underlined letters indicate residues substituted in Gαo-Nt[7]. Note that the expression of the KASH-construct carrying zDHHC11 results in the strong recruitment of Gαo-Nt[7]-GFP (**b**) and MGNC-Nt[7]-GFP (**d**) at the ONM, effect not seen by mRFP-KASH (**a**, **c**). Mean fluorescence intensity ratios of GFP-fusions at the ONM versus cytosol (**e**). Box plots indicate median (middle line), 25th, 75th percentile (box), and lowest, highest value (whiskers); three independent experiments (Gαo-Nt[7] and mRFP-KASH, $n = 44$; Gαo-Nt[7] and zDHHC11, $n = 45$; MGNTC-Nt[7] and mRFP-KASH, $n = 49$; MGNTC-Nt[7] and zDHHC11, $n = 43$). **f–k** The SwissKASH assay using zDHHC3 (**f**, **h**) and zDHHC7 (**g**, **i**). Gαo-Nt[7]-GFP efficiently targets the ONM by the co-expression of mRFP-zDHHC3-KASH (**f**) and mRFP-zDHHC7-KASH (**g**). MGNC-Nt[7]-GFP poorly localizes at the ONM in the presence of mRFP-zDHHC3-KASH (**h**) and mRFP-zDHHC7-KASH (**i**). Selected areas are magnified to the right. Mean fluorescence intensity ratios of GFP-fusions at the ONM versus cytosol by zDHHC3 (**j**) and zDHHC7 (**k**). Box plots indicate median (middle line), 25th, 75th percentile (box), and lowest, highest value (whiskers); three independent experiments (Gαo-Nt[7] and mRFP-KASH, $n = 44$; Gαo-Nt[7] and zDHHC3, $n = 50$; Gαo-Nt[7] and zDHHC7, $n = 46$; MGNTC-Nt[7] and mRFP-KASH, $n = 49$; MGNTC-Nt[7] and zDHHC3, $n = 53$; MGNTC-Nt[7] and zDHHC7, $n = 45$). **l–o** ONM recruitment of Gαo-Nt[7]-GFP is not observed in the SwissKASH assay using the inactive DHHS-mutants for zDHHC3 (zDHHS3; **l**), zDHHC7 (zDHHS7; **m**), and zDHHC11 (zDHHS11; **n**). The zDHHS11 mutant shows no ONM recruitment of MGNC-Nt[7]-GFP (**o**). Boxed areas are zoomed-in to the right. **a–d**, **f–i**, **l–o** Scale bars, 10 μm. *P* values were determined using one-way ANOVA Šídák test for (**e**, **j**, **k**). ns: not significant. Source data are provided as a Source Data file.

construct of zDHHC5 but not zDHHC9 (Supplementary Fig. 8a–d). This effect was not seen when the DHHS-mutant of zDHHC5 was used, suggesting that GCP16/Golga7 is a substrate for zDHHC5 (Supplementary Fig. 8e–g). On the other hand, Gαo-Nt⁷ and MGNC-Nt⁷ were not targeted to the ONM in the SwissKASH system using GCP16/Golga7 together with zDHHC5 or 9 (Supplementary Fig. 8a–d).

We then tested a Flag-tagged Golga7b construct in the SwissKASH assay for zDHHC5 and also observed its accumulation at the ONM (Fig. 6a, b). Surprisingly, we also observed ONM recruitment of MGNC-Nt⁷-GFP but not Gαo-Nt⁷-GFP under these conditions (Fig. 6a, b); both Golga7b and MGNC-Nt⁷ were not targeted to the ONM when the DHHS-mutant of zDHHC5 was used (Fig. 6c). We were not able to detect any ONM localization of MGNC-Nt⁷ without the co-expression of Golga7b (Supplementary Fig. 5e), implying that the zDHHC5-Golga7b complex might be involved in substrate recognition similar to zDHHC9-GCP16/Golga7[45]. Quantification showed a significant ONM accumulation of MGNC-Nt⁷ over Gαo-Nt⁷ and control (Fig. 6d), which is not due to a differential protein-interaction with mRFP-zDHHC5-KASH, as both Nt⁷ constructs did not co-precipitate the PAT (Supplementary Fig. 8h).

Then, we asked whether the higher PM targeting of Gαo-Nt³¹ compared to Gαo-Nt⁷ might also be the outcome of different zDHHC substrate specificities. Thus, we tested if zDHHC3, 7, and 11 target Gαo-Nt³¹ to the ONM with the same efficiency than Gαo-Nt⁷. Although zDHHC11-KASH efficiently relocalized Gαo-Nt³¹-GFP to the ONM, zDHHC3 and zDHHC7 showed a weaker effect when compared to Gαo-Nt⁷-GFP (Fig. 6e–i). In addition, we did not observe ONM targeting of Gαo-Nt³¹ by the co-expression of mRFP-zDHHC5-KASH and Golga7b (Supplementary Fig. 8i). Therefore, the weaker activity of the Golgi-associated zDHHC3 and 7 toward Gαo-Nt³¹ predicts an increase in its PM accumulation upon time, in agreement with the experimental findings (Fig. 1d).

Together, our data suggest that the distinct steady-state localizations of Gαo-Nt⁷ and MGNC-Nt⁷ are due to the differential substrate specificities of PM- and Golgi-localized enzymatically active zDHHCs, which can further be differently influenced by the accessory proteins.

**Compartment-specific S-palmitoylation of Nt⁷.** The notion that compartmentalization is controlled by PATs implies that the localization of Nt⁷ constructs might be shifted toward the PM or Golgi by manipulating the expression level of specific zDHHCs. To test this hypothesis, we first co-transfected N2a cells with Gαo-Nt⁷-GFP and HA-zDHHC11, and determined its relative PM content and co-localization with the MannII-BFP Golgi marker. In addition, we co-expressed MGNC-Nt⁷-GFP together with HA-zDHHC3 or HA-zDHHC7. Remarkably, the PM localization of Gαo-Nt⁷ strongly increased by the overexpression of zDHHC11, whereas its co-localization with MannII-BFP showed a significant decrease (Fig. 7a–d and Supplementary Fig. 9a). On the other hand, MGNC-Nt⁷ presented a higher co-localization with the Golgi marker but a significant decrease in PM content by the co-expression of zDHHC3 or zDHHC7 (Fig. 7e–l Supplementary Fig. 9b). The effects of the different zDHHCs on Gαo-Nt⁷ and MGNC-Nt⁷ were not evident upon co-expression of the corresponding DHHS-mutants (Fig. 7b, f, j). The changes in the localization of Nt⁷ constructs were not due to variations in their expression level upon co-expression of the different zDHHCs (Supplementary Fig. 9c–f). No changes in the overall localization of Gαo-Nt⁷ and MGNC-Nt⁷ could be observed upon co-expression of zDHHC3/7 and zDHHC11, respectively (Supplementary Fig. 9g–i). Similarly, co-expression of zDHHC5 and

Golga7b appeared not to alter the localization pattern of the Nt⁷ constructs (Supplementary Fig. 9j, k). Intriguingly, [³H]palmitate metabolic labeling of Gαo-Nt⁷-GFP was increased exclusively by the co-expression of zDHHC3 and 7 (Fig. 7m, n), whereas only zDHHC11 and the zDHHC5-Golga7b combination rose the labeling of MGNC-Nt⁷-GFP (Fig. 7m, o). Thus, drastic changes in the localization of the Nt⁷ constructs seem not to be followed by significant variations in their overall S-palmitoylation status.

Next, we analyzed the localization of Gαo-Nt⁷ and MGNC-Nt⁷ in cells after depletion of zDHHCs. We employed HeLa instead of the murine N2a cells for this analysis, as siRNAs have been used successfully to downregulate the expression of human zDHHCs in this cell line[47]. We first determined if, in addition to zDHHC5 (Fig. 4c), zDHHC3, 7 and 11 show localizations in HeLa similar to those in N2a cells. Accordingly, HA-tagged zDHHC3 and 7 were predominantly localized at the Golgi, while zDHHC11 was at the PM (Supplementary Fig. 10a–c). Then, we expressed Gαo-Nt⁷-GFP or MGNC-Nt⁷-GFP in cells pretreated with siRNAs against the pairs of zDHHC3/7 and zDHHC5/11. Gαo-Nt⁷ and MGNC-Nt⁷ localized normally in HeLa cells transfected with control siRNA, with Gαo-Nt⁷ predominantly at the Golgi and MGNC-Nt⁷ mainly at the PM (Fig. 8a, d). Remarkably, the simultaneous depletion of the Golgi-associated zDHHC3 and 7 clearly affected the localization of Gαo-Nt⁷: the construct exhibited a much weaker presence at the Golgi whereas its targeting to the PM appeared enhanced (Fig. 8b). MGNC-Nt⁷, however, seemed not affected by this treatment (Fig. 8e), but its localization was notoriously shifted toward the Golgi in HeLa cells upon the concomitant downregulation of the PM-associated zDHHC5 and 11 (Fig. 8f). On the other hand, the distribution of the Gαo-Nt⁷ construct did not show major changes in cells depleted of zDHHC5 and 11 (Fig. 8c). Quantification showed a significant reduction in the relative Golgi localization of Gαo-Nt⁷ by the downregulation of zDHHC3/7, and a significant increase in the Golgi content of MGNC-Nt⁷ upon zDHHC5/11 depletion (Fig. 8g). Accordingly, [³H]palmitate metabolic labeling of Gαo-Nt⁷ and MGNC-Nt⁷ was strongly reduced by the simultaneous downregulation of zDHHC3/7 and zDHHC5/11, respectively (Fig. 8h, i). A lower but significant reduction in labeling was additionally observed for Gαo-Nt⁷ by the combined depletion of zDHHC5/11, and for MGNC-Nt⁷ by zDHHC3/7 (Fig. 8h, i), demonstrating that these closely related sequences are differentially palmitoylated by PM- and Golgi-associated zDHHCs.

Collectively, these data confirm the conclusion that local S-palmitoylation plays a pivotal role in the compartmentalization of Gαo-Nt⁷ and MGNC-Nt⁷, and that the localization pattern can be shifted upon the up- or downregulation of different PATs.

**Targeting full-length Gαo to the ONM.** We analyzed next if the SwissKASH assay might reveal local S-palmitoylation of the full-length Gαo. Thus, we expressed Gαo-GFP together with the KASH-fusions of zDHHC3, 7, and 11, as they showed activity toward Gαo-Nt⁷. Remarkably, full-length Gαo was targeted to the ONM by the zDHHC constructs (Fig. 9a–d). Alternatively, the Cys4 mutant of Gαo, MGNC-Gαo-GFP, localized at the ONM primarily when the KASH-fusion for zDHHC11 was co-expressed (Supplementary Fig. 10d), similar to what we observed for MGNC-Nt⁷ (Fig. 5d, h, i). MGNC-Gαo was additionally targeted to the ONM by co-expression of mRFP-zDHHC5-KASH and Golga7b, an effect not seen for Gαo (Fig. 9e, f). Overall, our data show that the distinct steady-state localization of Gαo is controlled by the PM- and Golgi-localized zDHHCs.

To explore a potential role of ER-resident PATs in Gαo localization, we co-expressed Gαo-GFP with the HA-tagged

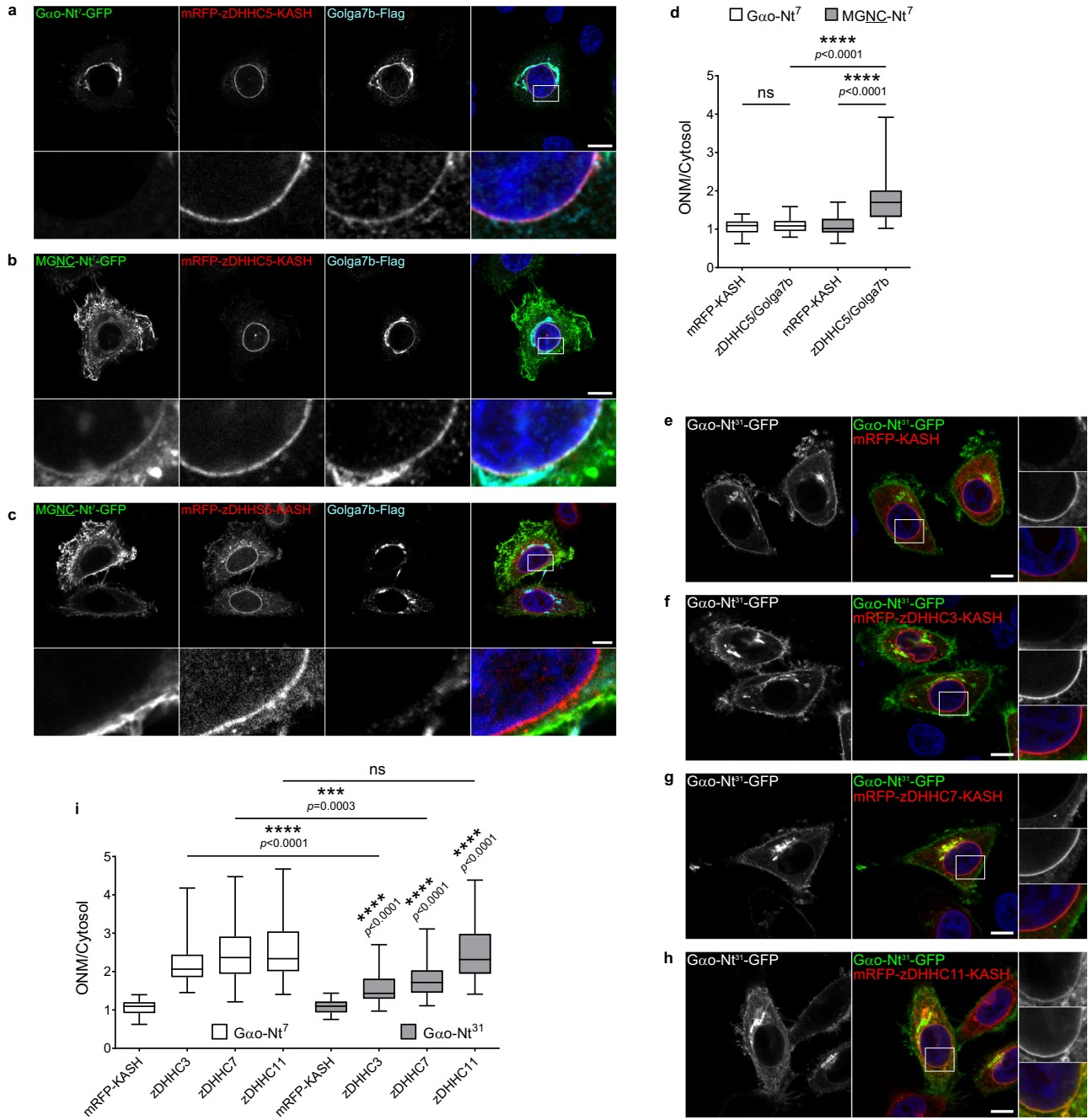

**Fig. 6 The zDHHC5-Golga7b complex. a–d** The SwissKASH assay for Gαo-Nt[7]-GFP (**a**) and MGNC-Nt[7]-GFP (**b**) using mRFP-zDHHC5-KASH and Golga7b-Flag in HeLa cells. Note that the KASH-fusion of zDHHC5 induces the ONM recruitment of MGNC-Nt[7]-GFP (**b**) and Golga7b (**a**, **b**), but not of Gαo-Nt[7]-GFP (**a**). ONM localization of MGNTC-Nt[7]-GFP and Golga7b is not induced by the inactive DHHS-mutant of zDHHC5 (zDHHS5; **c**). Cells were immunostained against Flag-tag and nuclei were stained in blue with DAPI. Marked regions are magnified at the bottom panels. Underlined letters indicate residues substituted in Gαo-Nt[7]. Mean fluorescence intensity ratios of GFP-fusions at the ONM versus cytosol (**d**). Box plots indicate median (middle line), 25th, 75th percentile (box), and lowest, highest value (whiskers); three independent experiments (Gαo-Nt[7] and mRFP-KASH, $n = 44$; Gαo-Nt[7] and zDHHC5-Golga7b, $n = 49$; MGNTC-Nt[7] and mRFP-KASH, $n = 49$; MGNTC-Nt[7] and zDHHC5-Golga7b, $n = 48$). **e–i** The SwissKASH assay for Gαo-Nt[31]-GFP using the control mRFP-KASH (**e**), zDHHC3 (**f**), zDHHC7 (**g**), and zDHHC11 (**h**). Color-channels are listed vertically top-to-bottom and selected areas are magnified to the right with the channels displayed also vertically in the same order. Mean fluorescence intensity ratio (ONM versus cytosol) of Gαo-Nt[31]-GFP compared to Gαo-Nt[7]-GFP (**i**). Box plots indicate median (middle line), 25th, 75th percentile (box), and lowest, highest value (whiskers); three independent experiments (For Gαo-Nt[7]: mRFP-KASH, $n = 44$; zDHHC3, $n = 50$; zDHHC7, $n = 46$; zDHHC11, $n = 47$. For Gαo-Nt[31]: mRFP-KASH, $n = 42$; zDHHC3, $n = 45$; zDHHC7, $n = 40$; zDHHC11, $n = 46$). **a–c**, **e–h** Scale bars, 10 μm. $P$ values were determined using one-way ANOVA Šídák test for (**d**, **i**). ns: not significant. Source data are provided as a Source Data file.

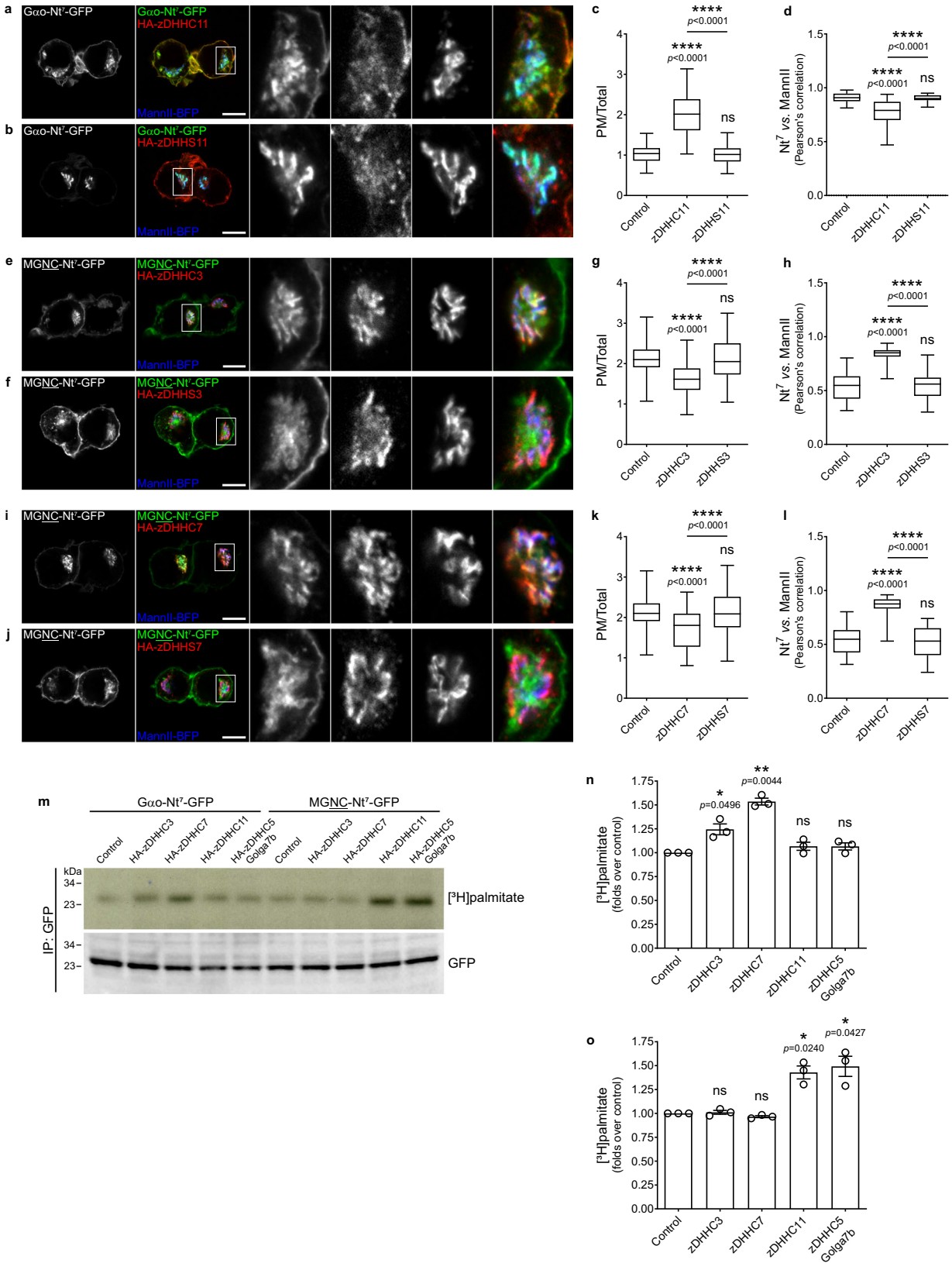

zDHHC1, 4, 6, 19, 20, and 24 in HeLa cells, but observed no apparent effect in Gαo distribution (Supplementary Fig. 10e). We noticed that zDHHC1 and to a lesser extend zDHHC20 were also present at the PM in HeLa cells (Supplementary Fig. 10e), a pattern that was not obvious in N2a cells (Supplementary Fig. 4b). Thus, we generated mRFP-KASH fusions for these PATs and

tested their ability to affect Gαo-Nt7 and MGNC-Nt7 localization in the SwissKASH assay. The KASH-fusion efficiently targeted zDHHC1 and 20 to the ONM, but they were not able to recruit Gαo-Nt7 or MGNC-Nt7 (Supplementary Fig. 11a–d).

Finally, we wondered whether the pool of Gαo targeting the ONM would keep its functionality. To address this, we first tested

**Fig. 7 zDHHC expression levels drive Nt[7] compartmentalization. a–d** Representative images of N2a cells expressing Gαo-Nt[7]-GFP (**a**, **b**) together with HA-zDHHC11 (**a**) or its inactive DHHS-mutants (HA-DHHS11; **b**). Golgi was labeled by the co-expression of MannII-BFP and HA-tagged zDHHCs were immunostained against HA (**a**, **b**). Color-channels are listed vertically top-to-bottom and selected areas are magnified with the channels displayed horizontally in the same order left-to-right. Mean fluorescence intensity ratio of Gαo-Nt[7]-GFP at the PM versus total cell (**c**), and co-localization with MannII-BFP (**d**). Box plots indicate median (middle line), 25th, 75th percentile (box), and lowest, highest value (whiskers); three independent experiments (For **c**: Control, $n = 58$; zDHHC11, $n = 61$; zDHHS11, $n = 54$. For **d**: Control, $n = 51$; zDHHC11, $n = 53$; zDHHS11, $n = 53$). **e–l** N2a cells expressing MGNC-Nt[7]-GFP (**e**, **f**, **i**, **j**) together with HA-zDHHC3 (**e**), HA-zDHHS3 (**f**), HA-zDHHC7 (**i**), or HA-DHHS7 (**j**). Underlined residues indicate substitutions in Gαo-Nt[7]. Mean fluorescence intensity ratio of MGNC-Nt[7]-GFP at the PM versus total cell (**g**, **k**), and co-localization with MannII-BFP (**h**, **l**). Box plots indicate median (middle line), 25th, 75th percentile (box), and lowest, highest value (whiskers); three independent experiments (For **g** and **k**: Control, $n = 56$; zDHHC3, $n = 53$; zDHHS3, $n = 56$; zDHHC7, $n = 54$; zDHHS7, $n = 52$. For **h** and **l**: Control, $n = 54$; zDHHC3, $n = 53$; zDHHS3, $n = 51$; zDHHC7, $n = 52$; zDHHS7, $n = 55$). **m–o** Representative [3H]palmitate metabolic radiolabeling of Gαo-Nt[7]-GFP and MGNC-Nt[7]-GFP upon the co-expression of HA-zDHHC3, HA-zDHHC7, HA-zDHHC11, or HA-zDHHC5 plus Golga7b-Flag. GFP-fusions were immunoprecipitated (IP), and analyzed by autoradiography ([3H]palmitate) and anti-GFP antibody (**m**). Radioactivity incorporated to Gαo-Nt[7] (**n**) and MGNC-Nt[7] (**o**) normalized to the respective controls. Data represent mean ± s.e.m.; three independent experiments. **a**, **b**, **e**, **f**, **i**, **j** Scale bars, 10 μm. *P* values were determined using one-way ANOVA Šídák test for **c**, **d**, **g**, **h**, **k**, **l** and one-sample *t*-test for (**n**, **o**). ns: not significant. Source data are provided as a Source Data file.

the ability of Gαo to interact with Gβ1γ3. As expected, Gαo-GFP and mRFP-Gβ1γ3 strongly co-localized at the PM and endomembranes in control HeLa cells (Supplementary Fig. 11e). When these G proteins were expressed together with zDHHC11 in the SwissKASH assay, Gβ1γ3 relocalized to the ONM in a pattern identical to Gαo (Fig. 9g). We tested next if a downstream signaling molecule might be co-recruited to the ONM by Gαo, and selected RGS19 which preferentially interacts with the active, GTP-loaded Gαo[48]. In addition to Gαo wild-type, we used the GTPase-deficient mutant of Gαo (Q205L) that is present in the GTP-loaded state in cells. Accordingly, a clear ONM accumulation of a His6-tagged RGS19 construct was observed when the SwissKASH system was applied to the Q205L mutant (Fig. 9h) but not the wild-type Gαo (Supplementary Fig. 11f). Altogether, our SwissKASH system shows that local S-palmitoylation of Gαo can cause its in-situ accumulation in the signaling-competent form, indicating that zDHHC selectivity and subcellular localization are key players in the substrate compartmentalization of the G protein, and potentially PMPs in general.

## Discussion

The fact that Gα subunits can be found at the Golgi apparatus is known for 30 years[49]. The Golgi localization of Gα subunits was assumed to be part of their trafficking pathway to the cell periphery, although some non-canonical functions of G-proteins were described at compartments other than the PM[50]. Only recently, the Golgi-pool of Gα subunits has been implicated in the downstream signaling of typical GPCRs[3] and the KDELR[4,5]. Thus, subcellular compartmentalization has emerged as an important player in G protein functions. The present study provides an in-depth characterization of key elements controlling N-myristoylation, S-palmitoylation and localization of Gαo. More importantly, we developed a system—SwissKASH—for the rapid detection of zDHHC-substrate pair relations in intact cells, opposing to the disruptive biochemical and proteomic techniques currently used to study S-palmitoylation[38]. Altogether, this work supports a model in which local S-palmitoylation by distinct zDHHCs is crucial for Gαo compartmentalization, and most likely for other PMPs (Fig. 10). In this model, N-myristoylation allows the nascent protein to reach the whole endomembrane system. Subsequently, specific PM- and Golgi-localized zDHHCs catalyze S-palmitoylation of PMPs, which in turn accumulate at these compartments. Our model is in agreement with the local S-palmitoylation described for substrates such as PSD-95 at dendritic spines[37] and calnexin at the ER[47], but adds the Golgi as a compartment in which PMPs can be locally modified and retained. To our knowledge, S-palmitoylation via the large number of Golgi-localized zDHHCs has been exclusively

discussed in the context of Golgi-to-PM transport of integral[51] and PMPs[12]. We hereby challenge this notion and show instead that PMP localization to one compartment or another (including Golgi and PM) is not a consequence of sequential trafficking of the PMPs but rather a direct localization driven by the substrate-specific PAT activities at different compartments.

The fact that zDHHCs can modify multiple substrates has led to the notion that S-palmitoylation is nonspecific and proximity-based. Accordingly, a recent study demonstrated the stochastic S-palmitoylation of engineered Cys residues in membrane proteins, proposing that catalysis is determined by substrate accessibility and not specific sequences[52]. Several zDHHCs, however, contain PDZ-binding motifs, SH3 domains or ankyrin-repeats outside the catalytic DHHC core that have been implicated in enzyme-substrate interactions[11]. Importantly, our study uncovered a striking substrate specificity for some zDHHCs toward peptides as small as Gαo-Nt[7]. We additionally showed that minor sequence modifications might result not only in drastic changes in substrate recognition by zDHHCs, but also in substrate localization (Fig. 10). Another interesting implication of this model is that cells and tissues could shift the localization—and associated function —of PMPs toward one specific compartment to another by controlling the expression of zDHHCs. The large number of zDHHC genes and their distinct tissue-specific expression patterns in humans is thus not surprising[34]. This property might be widespread among Metazoa as Nt[7] constructs showed remarkably similar localization in *Drosophila* and mammalian cells, and *Drosophila* contains also a large number of zDHHC genes that are differentially expressed in embryonic and adult tissues[53]. Regulation of compartmentalized PMP functions might also be acutely regulated by controlling the action of PATs locally. This seems true for zDHHC5, as recent studies point to an intricated regulation of its activity and localization by posttranslational modifications and accessory proteins (see below)[43,54].

The SwissKASH system we developed relies on tagging zDHHCs at their C-terminus with the KASH domain of syne-1/nesprin-1 for the ectopic ONM targeting. We applied this strategy only to PM- and Golgi-localized zDHHCs but we expect it will also work for the remaining PATs, excluding zDHHC24 with a predicted C-terminus facing the lumen[38]. Some of the ER-localized zDHHCs, however, showed a prominent ONM localization that might hinder their usage in the SwissKASH assay. In this case, alternative tags for the targeting of ER-localized zDHHCs to the PM, Golgi or mitochondria might be developed. The C-terminal domain of zDHHCs contains two non-catalytic TTxE and PaCCT (palmitoyltransferase conserved C-terminus) motifs that seem relevant for the overall protein structure[55]. The extreme C-terminal region of zDHHCs appears

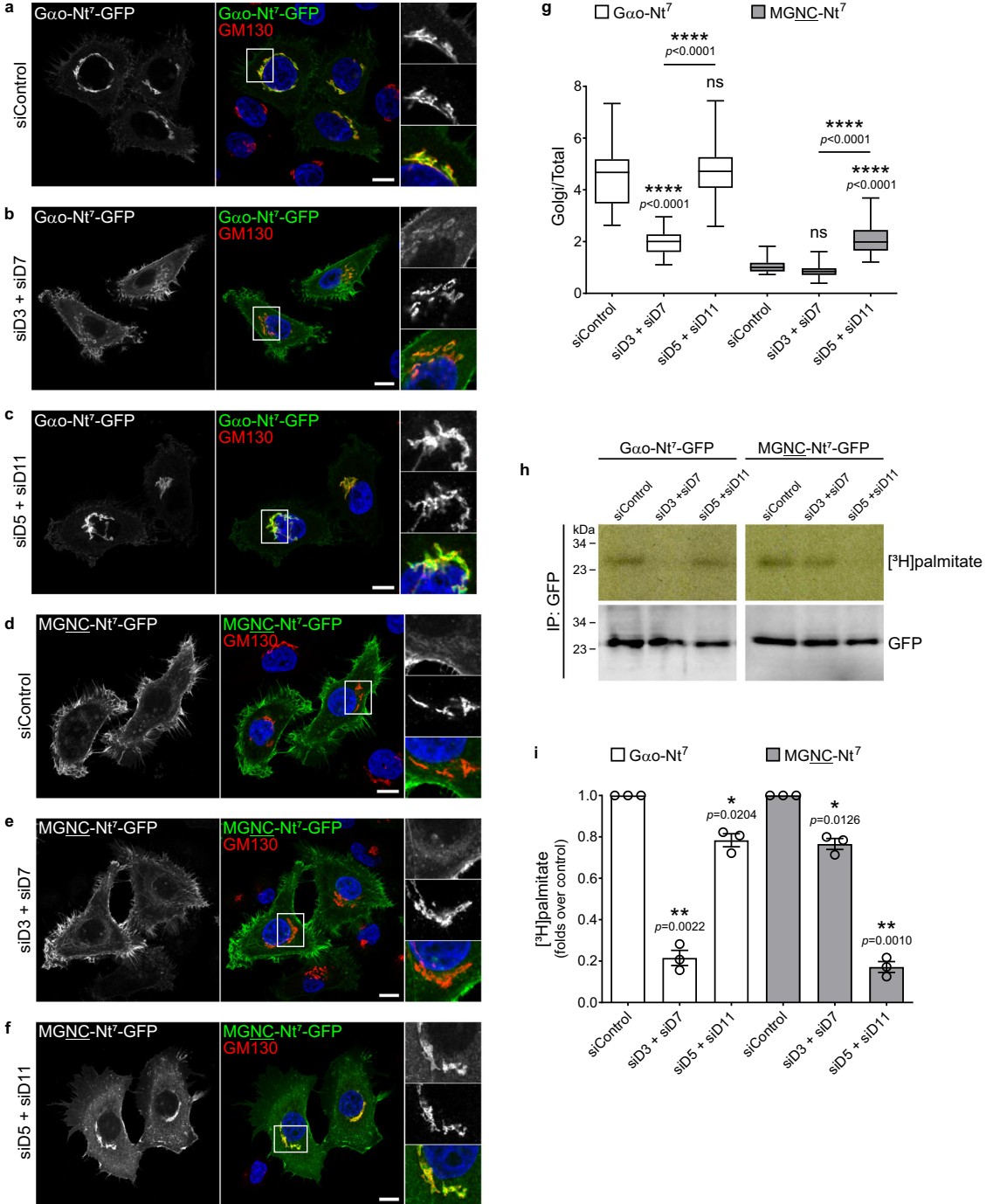

**Fig. 8 Downregulation of zDHHCs affects Nt[7] compartmentalization. a–g** Representative images of HeLa cells expressing Gαo-Nt[7]-GFP (**a–c**) or MGNC-Nt[7]-GFP (**d–f**). Cells were previously transfected with control siRNA (siControl; **a**, **d**), siRNAs against zDHHC3 and 7 (siD3 + siD7; **b**, **e**), or against zDHHC5 and 11 (siD5 + siD11; **c**, **f**). Cells were immunostained against GM130 to mark the Golgi, and nuclei were stained with DAPI in blue. Color-channels are listed vertically top-to-bottom and selected areas are magnified with the channels displayed also vertically in the same order. Underlined letters indicate residues substituted in Gαo-Nt[7]. Mean fluorescence intensity ratios of GFP-constructs at the Golgi vs total cell (**g**). Box plots indicate median (middle line), 25th, 75th percentile (box), and lowest, highest value (whiskers); three independent experiments (For Gαo-Nt[7] (**a–c**): siControl, $n = 57$; siD3 + siD7, $n = 57$; siD5 + siD11, $n = 60$. For MGNC-Nt[7] (**d–f**): siControl, $n = 55$; siD3 + siD7, $n = 58$; siD5 + siD11, $n = 57$). **h–i** Representative [³H] palmitate metabolic radiolabeling of Gαo-Nt[7]-GFP and MGNC-Nt[7]-GFP from cells pretreated with siControl, siD3 + siD7, and siD5 + siD11. GFP-fusions were immunoprecipitated (IP), and analyzed by autoradiography ([³H]palmitate) and anti-GFP antibody (**m**). Radioactivity incorporated to Gαo-Nt[7] and MGNC-Nt[7] normalized to siControl. Data shown as the mean ± s.e.m.; three independent experiments. **a–f** Scale bars, 10 μm. *P* values were determined using one-way ANOVA Šídák test for (**g**) and one-sample *t*-test for (**i**). ns: not significant. Source data are provided as a Source Data file.

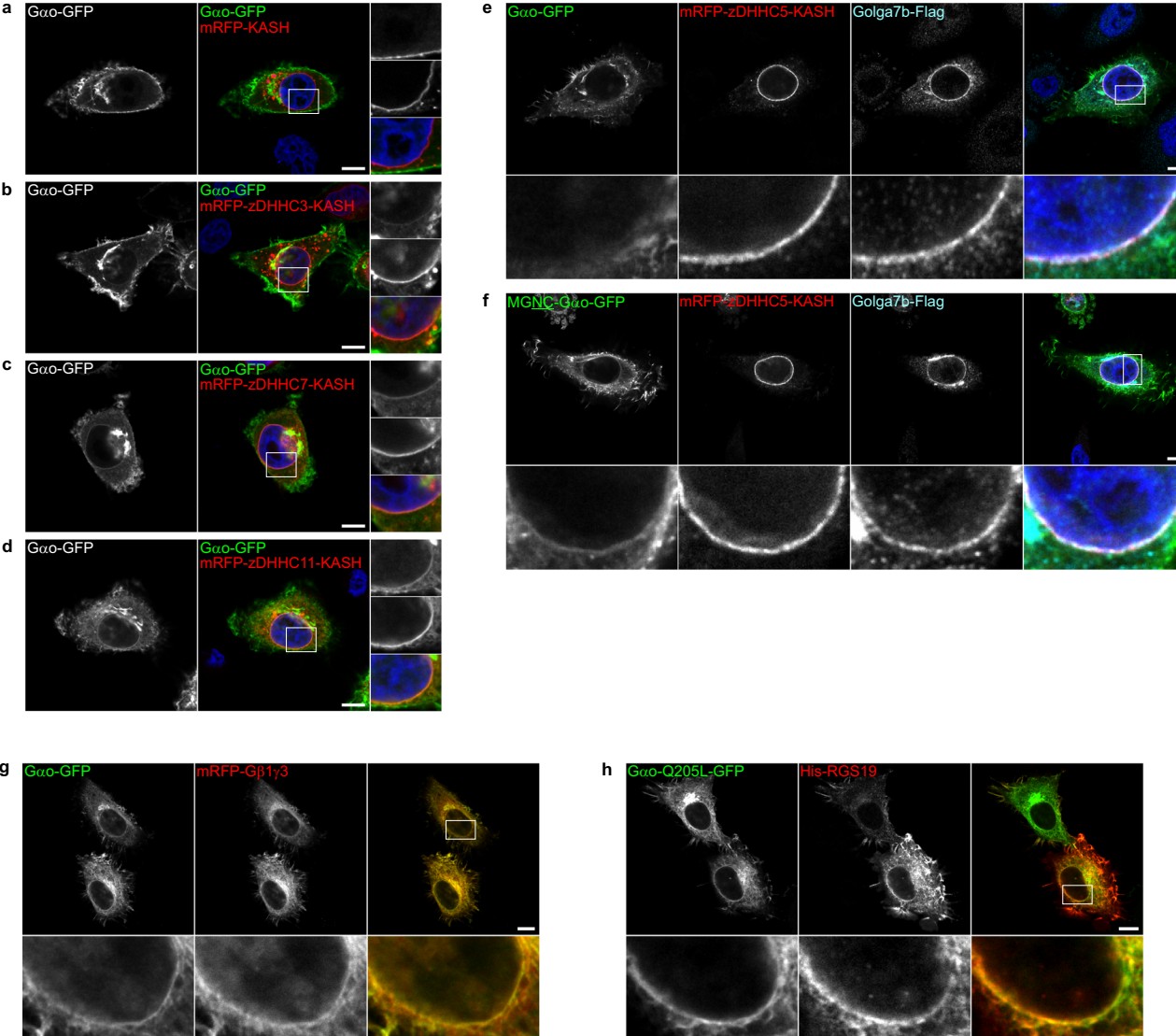

**Fig. 9 Targeting of full-length Gαo to the ONM. a–d** The SwissKASH assay for full-length Gαo (Gαo-GFP) using the control mRFP-KASH (**a**), zDHHC3 (**b**), zDHHC7 (**c**) and zDHHC11 (**d**) in HeLa cells. Nuclei stained in blue with DAPI. Color-channels are listed vertically top-to-bottom and selected areas are magnified to the right with the channels displayed also vertically in the same order. Note that the ONM accumulation of Gαo-GFP is induced by all zDHHCs but not the control mRFP-KASH. **e, f** The full-length Gαo-GFP (**e**) and its MGNC mutant (MGNC-Gαo-GFP; **f**) were tested in the SwissKASH system using mRFP-zDHHC5-KASH and Golga7b-Flag. Cells were immunostained against Flag-tag. Marked regions are magnified at the bottom panels. Underlined letters indicate residues substituted in the Nt[7] region of Gαo. Note that only MGNC-Gαo-GFP (**f**) and Golga7b (**e, f**) accumulate at the ONM. **g, h** The SwissKASH assay applied to Gαo-GFP (**g**) and its GTPase-inactive mutant Q205L (**h**) together with mRFP-Gβ1γ3 (**g**) or His$_6$-tagged RGS19 (His-RS19; **h**), and using BFP-zDHHC11-KASH (not displayed). Cells were immunostained against the His$_6$-tag (**h**). Selected areas are magnified at the bottom panels. **a–h** Scale bars, 10 μm.

unstructured and is most likely not involved in the binding and presentation of substrates[11], implying that the fusion of a KASH domain does not interfere with substrate specificity.

The SwissKASH pointed to zDHHC3, 7, and 11 as the major PATs for Gαo. This result agrees with a previous study showing zDHHC3 and 7 as the main PATs for Gαi2, Gαq and Gαs[56], while zDHHC11 activity toward Gα subunits has not been detected. Similarly, we were unable to detect an increase in [³H]palmitate labeling of Gαo-Nt[7] by the overexpression of zDHHC11, although it significantly changed Gαo-Nt[7] localization pattern. This suggests that the low starting level of S-palmitoylation of Gαo-Nt[7] at the PM makes it particularly sensitive to overexpression of zDHHC11, sufficient to relocalize significant quantities of Gαo-Nt[7] to this compartment without a measurable increase in [³H]palmitate. Reciprocally, the low starting levels of

MGNC-Nt[7] at the Golgi make this construct to significantly relocalize to this compartment upon overexpression of zDHHC3/ 7—again without accompanying measurable [³H]palmitate increase. In contrast, siRNA-mediated downregulations of zDHHCs result in stronger changes in the overall S-palmitoylation, thus confirming that these closely related Nt[7] sequences are differentially modified by PM- and Golgi-associated PATs. The SwissKASH system, therefore, appears more sensitive than the traditional metabolic labeling approach, as even small amounts of a substrate are visible at the ONM.

Several recent studies have revealed that various zDHHCs are regulated by accessory proteins which control their activity, stability, and/or localization[43]. We focused here on GCP16/Golga7 and Golga7b, and introduced them into the SwissKASH approach. We found that the zDHHC5-Golga7b complex

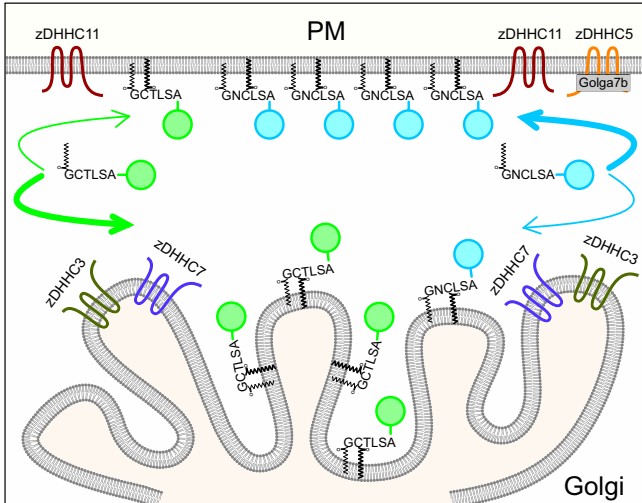

**Fig. 10 Model of compartmentalization of PMPs via local S-palmitoylation.** Model of how some peripheral membrane proteins might achieve specific subcellular compartmentalization by the interplay of the substrate selectivity of different zDHHCs. The model uses Gαo-Nt[7]-GFP (green circles) and MGNC-Nt[7]-GFP (cyan circles) as examples.

efficiently targeted MGNC-Nt[7] to the ONM, an effect not observed for Gαo-Nt[7] or in the absence of Golga7b. Interestingly, our MGNC-Nt[7] construct resembles the N-terminus of flotillin-2 (also known as reggie-1) that is myristoylated at Gly2 and palmitoylated at Cys4 by zDHHC5[42], and it is similarly localized at the PM but excluded from Golgi[57]. These data indicate that the position of the palmitoylatable Cys within a myristoylated Nt[7] peptide might be relevant for substrate specificity by zDHHC5. Moreover, zDHHC5 efficiently targeted both GCP16/Golga7 and Golga7b to the ONM although only the zDHHC5-Golga7b complex co-recruited MGNC-Nt[7], pointing to striking different functions for these complexes. Thus, the SwissKASH system might also be applied for functional studies of the complexes formed by zDHHCs and accessory proteins.

A polybasic stretch present in the N-terminal region of all Gα subunits has been implicated in PM binding[58]. The Gαo-Nt[31] construct containing the basic region within the α-helix showed indeed a greater steady-state localization at the PM compared to Gαo-Nt[7]. In the SwissKASH system, the presence of the α-helix in Gαo-Nt[31] appears to impact substrate recognition by the Golgi-associated zDHHC3 and 7, but not by the PM-localized zDHHC11. Thus, these data point to a second level of complexity in substrate affinity by zDHHCs, where regions distant from the palmitoylatable Cys might subserve regulatory functions. How the polybasic region in Gαo impacts the substrate specificity by zDHHCs remains to be clarified.

S-palmitoylation has gained much attention lately as a potential target for drug discovery to combat pathologies mediated by proteins such as the oncogenic Ras and infectious diseases like malaria caused by *Plasmodium falciparum* or the respiratory syndromes caused by coronaviruses[59–61]. Our SwissKASH system could enable screens for molecules that perturb zDHHC-specific S-palmitoylation of pathological proteins. This approach may provide an attractive alternative to directly target the desired oncogene (or protein of another pathogenic activity), focusing instead on developing tools to target the zDHHCs controlling the proper (or pathological) localization of the protein. In this regard, zDHHCs can be targeted by the drug discovery efforts. The large number of these enzymes in humans and their distinct expression patterns provide the reasoning to expect the power of such an approach in obtaining the favorable pharmacodynamics with

limited side effects. Another approach might aim at targeting the specific zDHHC-substrate interaction pairs; this avenue is expected to deliver even more specific targeted therapies.

To sum up, we have delineated the code describing the Golgi vs. PM localization of the G protein Gαo, showed that these two localizations are independent from the previously assumed Golgi-to-PM trafficking, and discovered a general mechanism controlling PMP localization to its target membrane through the substrate selectivity of differently localized PATs. This major insight into the cell biology of protein subcellular localization may pave the way to drug discovery principles.

## Methods

**Antibodies and reagents.** Primary antibodies (Abs) for immunofluorescence (IF) and Western blots (WBs): monoclonal Abs (mAbs) anti-flotillin-2 (610383; WB: 1/1000) and anti-GM130 (610823; IF: 1/500) from BD Biosciences, mAb anti-HA-tag (11867423001; IF: 1/500, WB: 1/2000) from Roche, mAb anti-mRFP/DsRed2 (sc-101526; WB: 1/250) from Santa Cruz Biotechnology, mAb anti-His₆ (34650; IF: 1/500) from Qiagen, mAb anti-GAPDH (GTX28245; WB: 1/2000) and polyclonal antibody (pAb) anti-GFP (GTX113617; WB: 1/2000) were from GeneTex, mAb anti-α-tubulin (T6199; IF: 1/4000) and pAb anti-Flag-tag (F7425; IF: 1/500, WB: 1/1000) were from Sigma-Aldrich, and a pAb anti-GFP (632592; WB: 1/1000) was from Takara. All secondary Abs for immunofluorescence (IF) and Western blots (WBs) were from Jackson ImmunoResearch: anti-Mouse Alexa Fluor 488-conjugated (111-545-144; IF: 1/500), anti-Mouse Cy3-conjugated (115-165-146; IF: 1/500), anti-Rat Cy3-conjugated (112-165-143; IF: 1/500), anti-Rabbit Cy5-conjugated (111-175-144; IF: 1/500), anti-Mouse Horseradish peroxidase (HRP)-conjugated (115-035-062; WB: 1/5000), and anti-Rabbit HRP-conjugated (111-035-144; WB: 1/5000). DAPI (32670), Cycloheximide (C4859), and Palmostatin B (178501) were from Sigma-Aldrich, D/D solubilizer (635054) from Takara, VECTASHIELD Mounting Medium (H-1400) from Vector Laboratories, Glutathione Sepharose 4B beads (17075601) from Cytiva, DDD85646 (13839) from Cayman Chemical, 2-bromopalmitate (sc-251714) from Santa Cruz Biotechnology, and Membrane lipid strips (P-6002) from Echelon Biosciences.

**Cell lines and culture conditions.** Male mouse neuroblastoma Neuro-2a (N2a; ATCC, CCL-131) and female human epithelial HeLa (ATCC, CCL-2) cells were maintained in MEM (Thermo Fisher Scientific), supplemented with 10% FCS, 2 mM L-glutamine, 1 mM pyruvate, and 1% penicillin-streptomycin at 37 °C and 5% $CO_2$. Male *Drosophila* Schneider-2 (S2; Invitrogen, R690-07) cells were maintained in Schneider's Drosophila Medium (Lonza) supplemented with 10% FCS and 1% penicillin-streptomycin at 28 °C. All vector transfections were carried out with X-tremeGENE HP (Roche, XTGHP-RO) or FuGENE HD (Promega, E2311) according to the manufacturer's instructions.

**Plasmids and molecular cloning.** The plasmids Gαo-GFP WT and Q205L mutant, GalT-GFP, His₆-RGS19, mRFP-Gβ1, mRFP-Gγ3, and GalT-mRFP for *Drosophila* expression were previously described[5,48,62]. To generate the GFP-fusion of Gαo-Nt[7] and Gαo-Nt[31], the fragments were PCR-amplified (primers listed in Supplementary Data 3) from Gαo-GFP, products were cut with KpnI/AgeI and cloned in frame into the same sites of pEGFP-N1 (Clontech). The Gαo-Nt[8–31]-GFP sequence was PCR-amplified from Gαo-Nt[31]-GFP, the product cut with KpnI/NotI and used to replace the GFP sequence in pEGFP-N1 cut with the same enzymes. Gαo-Nt[7]-GFP was used as template for the PCR amplification of the following Nt[7]-GFP sequences: Gαo-G2L, Gαo-C3N, Gαo-S6A, Gαo-S6C, Gαo-S6G, Gαo-S6T, Gαo-S6F, Gαo-S6N, Gαo-S6R, Gαo-S6V, MGNC, MGNTC, MGCKR, MGCKE, MGCDR, MGCDE, MGCKH, MGCDH, MGSLCSR, MGSSCSR, and MGLLCSR. All PCR products were cut with KpnI/NotI and then used to exchange the corresponding GFP sequence from pEGFP-N1. For *Drosophila* expression, the plasmids containing the constructs Gαo-Nt[7]-GFP, MGNC-Nt[7]-GFP, and MGSLCSR-Nt[7]-GFP were cut with EcoRI/NotI and inserts were ligated into the same sites of pAc5.1/V5-HisA (Thermo Fisher Scientific, V411020). The full-length Gαo mutants MGNC-Gαo, Gαo-C3N and MGSLCSR-Gαo were PCR-amplified from Gαo-GFP, cut and inserted in frame into the KpnI/ApaI sites of pEGFP-N1. For the generation of Gαo-Nt[7]-FM[4]-GFP and MGNC-Nt[7]-FM[4]-GFP, the FM[4] sequence was PCR-amplified from the GFP-FM[4]-hGH plasmid (Andrew Peden, University of Sheffield), cut with AgeI/PstI and ligated in frame into the same sites of Gαo-Nt[7]-GFP and MGNC-Nt[7]-FM[4]-GFP. The mRFP-Lact-C2 was done by replacing the AgeI/BsrGI GFP sequence from GFP-Lact-C2 (kindly provided by Gregory Fairn; University of Toronto) with the sequence from pmRFP-C1. MannII-mRFP was created by exchanging the BamHI/NotI sequence from MannII-BFP[63] with the equivalent sequence from pmRFP-N1 (Claudia Stuermer, University of Konstanz). The mRFP-KASH and BFP-KASH plasmids were cloned by substituting the AgeI/BsrGI GFP sequence in the GFP-KASH plasmid[64] with the analogous sequence from pmRFP-C1 and pEBFP2-C1 (Addgene, 54665), respectively. The plasmid for the non-tagged SUN2 expression was cloned by cutting the SUN2-CFP plasmid[65] with EcoRI and BamHI, blunting with Phusion DNA Polymerase (New England Biolabs, M0530), and re-ligating the plasmid thus

introducing a stop codon in frame with the SUN2 sequence. The SUN2-GFP was generated by replacing the SalI/NotI CFP fragment with the XhoI/NotI GFP fragment from pEGFP-N1. The mRFP-zDHHC5 wild-type and DHHS-mutant were cloned by cutting the BamHI/NotI sequence of pCI-neo-Flag-DHHC5 and pCI-neo-Flag-DHHS5[66], and ligating in frame into the BglII/PspOMI sites of pmRFP-C1. For the generation of mRFP-zDHHC5-KASH wild-type and DHHS-mutant, the sequences mRFP-zDHHC5 and mRFP-zDHHS5 were PCR-amplified from mRFP-fusions, cut with NheI/KpnI and used to exchange the GFP sequence cut with the same enzymes from GFP-KASH. Masaki Fukata (National Institutes of Natural Sciences, Japan) generously provided a collection of 23 mouse zDHHC isoforms cloned into the pEF-Bos-HA plasmid[36]. Slightly different cloning strategies were used for the generation of all mRFP-zDHHC-KASH constructs. Specifically, the zDHHC2, 11 and 18 sequences were PCR-amplified from the corresponding pEF-Bos-HA plasmids, cut with AgeI/KpnI and ligated in frame into the BspEI/KpnI sites of mRFP-KASH and also BFP-KASH for zDHHC11. The sequence for zDHHC1, 3, 7, 12, 13, 15, 16, 20, 21, and 25 were PCR-amplified from the original pEF-Bos-HA plasmids, cut with BglII/EcoRI and inserted into the same sites of mRFP-KASH. The zDHHC8 sequence was PCR-amplified from pEF-Bos-HA-DHHC8, cut with XhoI/EcoRI and ligated into the same sites of mRFP-KASH. zDHHC9 was PCR-amplified from pEF-Bos-HA-DHHC9, cut with SacI/BamHI and ligated into the same sites of mRFP-KASH. The zDHHC14 sequence was PCR-amplified from pEF-Bos-HA-DHHC14, cut with AgeI/BamHI and ligated into the BspEI/BamHI sites of mRFP-KASH. The zDHHC17 and 23 were PCR-amplified from the corresponding pEF-Bos-HA plasmids, cut with BglII/HindIII and inserted into the same sites of mRFP-KASH. All inactive DHHS-mutants for the relevant mRFP-zDHHC-KASH and HA-zDHHC constructs were obtained by point mutagenesis. The GFP-fusion of reggie-1/flotillin-2 was previously described[67]. Tamas Balla (National Institutes of Health) kindly provided the FAPP1-PH-GFP plasmid, Bo van Deurs (University of Copenhagen) the GFP-caveolin-1 construct, Scott Dixon (Stanford University) the GCP16/Golga7-Flag, and Mark Collins (University of Sheffield) the Golga7b-Flag. The GFP-SNAP23[68] and pGEX6P1-GFP-Nanobody[69] plasmids were obtained from Addgene.

**Structure alignment**. The Gαo structures 6g79, 6oik, and 6k41 were aligned using the PyMOL v2.3 (pymol.org). The N-terminal α-helix of 6g79 was set as reference to align all structures. A similar alignment was done with the Gαi1 structures 1gg2, 5kdo, 6ddf, 6osa, 6n4b, 6kpf, and 6k42, setting the N-terminal α-helix of 5kdo as reference. Publicly available Gα structures were obtained from RCSB (rcsb.org).

**Immunofluorescence and microscopy**. For microscopy, N2a and HeLa cells were transfected for 7 h, trypsinized, and seeded on poly-L-lysine-coated coverslips in complete MEM for an additional 15 h before fixation. When indicated, cells were seeded in complete MEM supplemented with 10 μM DDD85646 or 100 μM 2-bromopalmitate. S2 cells were transfected for 24 h, washed one time with PBS, resuspended in complete media, and seeded on poly-L-lysine-coated coverslips for 30 min before fixation. All cells were fixed for 20 min with 4% paraformaldehyde in PBS. For immunostaining, cells were permeabilized for 1 min using ice-cold PBS supplemented with 0.1% Triton X-100, blocked for 30 min with PBS supplemented with 1% BSA, incubated with the primary antibody in blocking buffer for 2 h at room temperature (RT), washed and subsequently incubated with secondary antibodies and DAPI in blocking buffer for 2 h at RT. Coverslips were finally mounted with VECTASHIELD on microscope slides. Cells were recorded with a Plan-Apochromat 63x/1.4 oil objective on a LSM800 Confocal Microscope using the ZEN 2.3 software (all Zeiss). When required, mean fluorescence intensity was determined from confocal images using ImageJ v1.53c (National Institutes of Health). Images were not recorded using the same confocal settings, therefore ratio fluorescence values, such as Golgi fluorescence vs. total fluorescence or PM vs. total, were used for quantifications (see below). As a proof of validity of this approach, we recorded N2a cells expressing Gαo-Nt[7]-GFP or Gαo-Nt[31]-GFP under identical confocal settings, and found that to a similar Golgi content of both constructs, Gαo-Nt[31] showed consistently higher PM values than Gαo-Nt[7] (Supplementary Fig. 11g). Taken values together, the Golgi signal from Gαo-Nt[7] is ~1.5-fold higher, and its PM signal is ~2-fold lower, than those of Gαo-Nt[31] (Supplementary Fig. 11h), recapitulating the ratio representation we adopted in Fig. 1e, f and further elsewhere. All images were finally edited using ZEN lite 3.3 (Zeiss) and CorelDRAW 2020 (Corel).

**PM and Golgi accumulation**. N2a cells expressing the GFP-fusion constructs were immunostained as indicated above using a mouse monoclonal antibody (mAb) against GM130 to visualize the Golgi apparatus, and a DAPI staining for the nucleus (both not displayed in all final images). Alternatively, co-expression of the MannII-BFP construct was used to label the Golgi (see below under "co-localization analysis" for plasmid ratio used for transfection). To avoid interferences due to different expression levels of GFP-constructs among cell population, mean fluorescence intensity was measured at the GM130- or MannII-BFP-positive Golgi region as well as at the total cell area, and ratio values were used to determine the relative Golgi accumulation of the constructs. Simultaneously, mean fluorescence intensity was determined at an unbroken region of the PM lacking membrane

protrusions, and the ratio over total cell fluorescence was used to define relative PM content for each GFP-construct.

**Co-localization analysis**. N2a cells were transfected and immunostained against GM130 as described above. To determine co-localization of the various GFP constructs and the GM130 signal, an area covering the whole perinuclear region but excluding the PM was selected in confocal images and the Pearson's correlation was calculated using the co-localization tool of ImageJ v1.53c. The same analysis was done in N2a cells co-expressing the phosphatidylserine biosensor mRFP-Lact-C2[25] and the different GFP-fusions, transfected at equal plasmid ratio. The Golgi marker GalT-GFP was used as control. This analysis was additionally done in N2a cells co-transfected with GFP-fusions, HA-tagged zDHHCs or empty pcDNA3.1(+), and MannII-BFP as the Golgi marker (2:4:1 plasmid ratio).

**siRNA knockdown experiments**. For siRNA transfection, $1.2 \times 10^5$ HeLa cells were seeded on culture plates, and transfected 24 h later with 40 pmol of control siRNA (1027281) or 20 pmol of each siRNAs against human zDHHC3 (SI02777642), zDHHC5 (SI04159694), zDHHC7 (SI00766850), and zDHHC11 (SI04365914) (all Qiagen) using TransIT-X2 transfection reagent (Mirus, MIR 6004). siRNAs against the different zDHHCs were previously validated in HeLa cells[47]. After 48 h, cells were transfected with GFP-fusion constructs, and 24 h later cells were fixed and immunostained against GM130. Golgi accumulation of the constructs was determined from confocal images (all steps as described above). The relative PM localization of GFP-constructs was not determined for HeLa cells due to their heterogenous morphology, which does not allow for the clear visualization of a PM monolayer in the majority of the cells within the population.

**Biochemical analyses**. Transfected N2a cells were lysed with ice-cold Lysis buffer (20 mM Tris-HCl, pH 7.5, 100 mM NaCl, 5 mM MgCl₂, 2 mM EDTA, 1% Triton X-100, 0.1% SDS, and 10% glycerol) supplemented with a protease inhibitor cocktail (Roche). Extracts were cleared by centrifugation at $15,000 \times g$ and 4 °C for 15 min, boiled at 95 °C for 5 min and finally analyzed by SDS-PAGE and Western blots using antibodies against GFP, α-tubulin as loading control, and HA-tag when required. HRP-conjugated secondary antibodies allowed for enhanced chemiluminescence (ECL) detection in a Fusion FX6 Edge system (Vilber). Quantification of all blots was done using ImageJ v1.53c, and images were edited using EvolutionCapt v18.11 (Vilber) and CorelDRAW 2020.

**Co-immunoprecipitation**. The recombinant GST-tagged Nanobody against GFP[69] expressed in *Escherichia coli* Rosettagami (Novagen, 71351) was purified with Glutathione Sepharose 4B beads according to the manufacturer's instructions. Protein purity was assessed by SDS-PAGE and Coomassie blue staining.

N2a cells were co-transfected with the different Gαo-GFP constructs and mRFP-Gβ1/Gγ3 at 1:1:1 plasmid ratio. HeLa cells were transfected with the constructs used for the SwissKASH assay at the same plasmid ratio described below. SUN2-GFP was used as positive control for the interaction with mRFP-zDHHC-KASH constructs. After 24 h transfection, cells were resuspended with ice-cold GST-lysis buffer (20 mM Tris-HCl, pH 8.0, 1% Triton X-100 and 10% glycerol in PBS) supplemented with a protease inhibitor cocktail (Roche) and 50 μM Palmostatin B (for SwissKASH-derived samples), and passed 10 times through a 25 G needle. Extracts were cleared by centrifugation at $15,000 \times g$ for 15 min at 4 °C, and supernatants were incubated with 2 μg of purified GST-tagged GFP-Nanobody for 30 min on ice. Then, 20 μL of Glutathione Sepharose 4B beads were added and samples rotated overnight at 4 °C. Beads were repeatedly washed with GST-lysis buffer, prepared for SDS-PAGE, and finally analyzed by Western blot using antibodies against GFP, mRFP, and Flag-tag when needed, as well as HRP-conjugated secondary antibodies for ECL detection. All co-immunoprecipitations were done in duplicate with very similar outcomes.

**Crude subcellular fractionation**. N2a cells were transfected for 24 h and cell extracts were prepared using non-denaturing conditions and fractionated by high-speed centrifugation[67]. Cells were rinsed twice with PBS and resuspended in hypo-osmotic buffer (20 mM HEPES, pH 7.0) supplemented with a protease inhibitor cocktail. The cell suspension was passed 20 times through a 25 G needle, and nuclei and unbroken cells were removed by centrifugation at $700 \times g$ for 10 min at 4 °C. The supernatant was centrifuged at $100,000 \times g$ for 60 min at 4 °C. The new supernatant (cytosolic fraction) was directly prepared for SDS-PAGE, and the pellet (membrane fraction) was gently washed with hypo-osmotic buffer and resuspended in GST-lysis buffer supplemented with 0.5% SDS and protease inhibitors. The membrane fraction was cleared by centrifugation at $15,000 \times g$ for 20 min at 4 °C, and prepared for SDS-PAGE. Western blot analysis was done as above using the antibodies against GFP, and flotillin-2 and GAPDH as endogenous membrane and cytosolic markers, respectively.

**Metabolic radiolabeling with [³H]palmitate**. [³H]palmitate radiolabeling was performed as previously reported[70]. Transfected HeLa cells were starved for 1 h in MEM supplemented with 10 mM HEPES, pH 7.4, and subsequently radiolabeled for 3 h with 70 μCi/ml of [9,10-³H]palmitate (American Radiolabeled Chemicals,

Inc., ART-0129-25). Alternatively, cells were starved and radiolabeled in the presence of 100 μM 2-bromopalmitate, or incubated in complete media with 50 μM Palmostatin B for 4 h previous to the starvation and radiolabeling, both done also in the presence of Palmostatin B. Cells were then washed, lysed in a 0.5 M Tris-HCl, pH 7.4 buffer containing 0.5% Nonidet P-40, 20 mM EDTA, 10 mM NaF, 2 mM benzamidine and a protease inhibitor cocktail (Roche), and cleared by centrifugation. Supernatants were incubated overnight with anti-GFP agarose beads (Chromotek, GTA-100), beads were then washed several times, and incubated for 10 min at room temperature with 1 M hydroxylamine, pH 7.4 (Sigma-Aldrich, 159417) or 2 M Tris-HCl, pH 7.4. Beads were finally prepared for and loaded into a 4–20% gradient SDS-PAGE. One third of the immnunoprecipitate was analyzed by WB using antibody against GFP, and two thirds were used for fluorography on film. Radiolabeled products were analyzed using a Typhoon Scanner v1.1.0.7 and the ImageQuant TL v8.1.0.0 software (Amersham). Metabolic labeling was done in triplicate with similar outcomes.

**Protein-lipid overlay assay**. After 24 h of transfection, N2a cells were washed twice with TBS (50 mM Tris-HCl, pH 8.0, 150 mM NaCl, and 10% glycerol) and cell extracts were prepared as explained above in TBS supplemented with 0.5% Tx-100 and protease inhibitors. Membrane lipid strips were blocked in 3% BSA in TBS supplemented with 0.1% Tween-20 (TBS-T) for 1 h at 4 °C. Then, strips were incubated overnight at 4 °C with cleared cell extracts previously diluted 1/10 in TBS-T supplemented with 1% BSA. Strips were incubated first with a pAb against GFP, and then with a secondary HRP-conjugated pAb to detect GFP-constructs bound to lipid dots. The protein-lipid overlay assay was done in duplicate for each condition with similar results.

**Reverse dimerization assay**. HeLa cells were co-transfected with the Golgi marker GalT-mRFP (not displayed in all figures) and Gαo-Nt⁷-FM⁴-GFP, MGNC-Nt⁷-FM⁴-GFP or GFP-FM⁴-hGH (at a 1:3 plasmid ratio), and seeded on poly-L-lysine-coated coverslips for 15 h as indicated above. Then, cells were incubated for 2 h at normal culture conditions in HBSS supplemented with 20 mM HEPES, pH 7.4, and 50 μM Cycloheximide to block de novo synthesis of proteins. The reverse dimerization was induced by adding fresh HBSS supplemented as above plus 1 μM D/D solubilizer for the time indicated in the corresponding figures. For the temperature block of Golgi transport, the last two steps were simultaneously performed at 37 and 20 °C. Then, cells were fixed and prepared for microscopy. Quantification of fluorescence intensity at the MannII-mRFP Golgi region was done as described above.

For the live imaging of reverse dimerization, HeLa cells were transfected as above but seeded on μ-Slide-4-wells coverslips (Ibidi). Cell were first incubated for 30 min at 37 °C in Hank's Balanced Salt Solution (HBSS; Gibco) supplemented with 20 mM HEPES, pH 7.4 and 50 μM cyclohexamide. Slides were then mounted on a temperature-controlled stage in a VisiScope CSU-X1 spinning disk confocal system (Visitron Systems) equipped with a Plan-Apochromat 63x/1.4 oil objective on an AxioObserver.Z1 microscope (Zeiss), a Evolve 512 EMCCD Camera (Photometrics), and the VisiView v4.00.10 Imaging software (Visitron Systems). Reverse dimerization was induced by adding D/D solubilizer to reach a 1 μM final concentration, and cells were immediately recorded at one image per 5 s for 10 min. For analysis, movies were generated from stacks using ImageJ v1.53c, and the mean fluorescence intensity of an area at the center of the MannII-mRFP Golgi region was measured from stacks.

**Palmostatin B treatment**. For live imaging, N2a cells co-expressing the GFP-fusion constructs and the MannII-mRFP Golgi marker were seeded on μ-Slide-4-wells coverslips (Ibidi). Cells were first incubated for 30 min at 37 °C in HBSS supplemented with 20 mM HEPES, pH 7.4 and 50 μM cycloheximide, then Palmostatin B in DMSO was added to a 50 μM final concentration, and cells were immediately recorded at one image per 30 s for 45 min in the spinning disk confocal system described above. Same volume of DMSO was added for control cells. PM content and co-localization with MannII-mRFP was done as above. For fixed samples, N2a cells expressing only GFP-fusions were treated as above but fixed after 45 min of Palmostatin B addition. Cells were then immunostained against GM130 to visualize the Golgi and DAPI for nuclei (not displayed in the figures).

**S-palmitoylation at the outer nuclear membrane (SwissKASH assay)**. HeLa cells were co-transfected as above with plasmids encoding the GFP-fusion, SUN2, and control mRFP/BFP-KASH or mRFP/BFP-zDHHC-KASH at a 1:3:3 plasmid ratio, at a 1:2:2:1 when GCP16/Golga7-Flag, Golga7b-Flag or His₆-RGS19 was included, or at 1:2:2:1:1 when mRFP-Gβ1 and mRFP-Gγ3 were co-transfected. Cells were then seeded on poly-L-lysine-coated coverslips, fixed and prepared for microscopy. For quantification, mean GFP-fluorescence intensity at the ONM region labeled by the mRFP-KASH fusion was determined using ImageJ 1.53c. Simultaneously, GFP-fluorescence was measured at a nearby cytosolic region as well, and the ratio ONM over cytosol was used to define the relative ONM content of each GFP-construct. Ratio values were used due to different expression levels of the GFP-constructs among the cell population.

**Statistics and reproducibility**. Statistical parameters, including the exact values of n are reported in figures and figure legends. Results in Box plots indicate median

(middle line), 25th, 75th percentile (box), and lowest, highest value (whiskers), other data are shown as the mean, and error bars represent the s.e.m. or s.d. as indicated in the corresponding figure legend; ns, not significant, *$p \leq 0.01$; **$p \leq 0.005$; ***$p \leq 0.001$; ****$p \leq 0.0001$ using one-way or two-way ANOVA test with Tukey or Šídák corrections (for multi-sample groups), two-sided unpaired t-test (for two-sample comparison), and one-sample t-test (for one-sample comparison to control). Prism 9 (GraphPad) was used to determine statistical significance. No statistical methods were used to predetermine the sample sizes. All replicates successfully reproduced the presented findings. No data were excluded from the analyses. The experiments were not randomized and investigators were not blinded to allocation during experiments and outcome assessment.

**Reporting summary**. Further information on research design is available in the Nature Research Reporting Summary linked to this article.

## Data availability

The crystal structure data used in this study are available in the RCSB PDB-database under accession codes 6G79, 6OIK, 6K41, 1GG2, 5KDO, 6DDF, 6OSA, 6N4B, 6KPF, and 6K42. The data generated in this study are provided in the Source Data, Supplementary Information, and Supplementary Data files. Source data are provided with this paper.

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

## Acknowledgements

The work was supported by the Swiss National Science Foundation grants 31003A_175658 to V.L.K. and 310030_192608 to F.G.V.D.G. We would like to thank Sabina Troccaz for the excellent technical assistance, Mikhail Kryuchkov for critical discussion, and members of the Bioimaging core facility of the CMU for assistance in microscopy.

## Author contributions

G.P.S. performed and analyzed experiments, and developed the methodologies. A.K. performed and quantified the Palmostain B live imaging. L.A. performed and analyzed [3H] palmitate metabolic labeling. J.V. assisted with protein structural analysis and molecular cloning. C.A. assisted with interpretation of results. F.G.V.D.G. analyzed [3H]palmitate metabolic labeling. G.P.S. and V.L.K. designed the work and wrote the paper. All authors reviewed the manuscript.

## Competing interests

The authors declare no competing interests.
