## [Peer Review File · Nature Communications]

Local and substrate-specific S-palmitoylation determines subcellular localization of GαREVIEWER COMMENTS

Reviewer #1 (Remarks to the Author):

The manuscript by Solis et al. investigated the role of “local S-palmitoylation” of Gαo in its subcellular compartmentalization. The authors developed the SpONM (S-palmitoylation at the outer nuclear membrane) assay to evaluate the substrate specificity of zDHHC PATs in cells, by ectopically targeting zDHHCs to the outer nuclear membrane. Using this assay, the authors showed the distinct substrate specificity of zDHHCs and identified zDHHC3 and 7 as the Golgi-localized PATs and zDHHC11 as the plasma membrane (PM)-localized PAT for Gαo-Nt7 proteins. Then, they showed that the expression level of the local zDHHCs PATs (Golgi vs PM) determines relative contents/distributions of Gαo between Golgi and PM. Their proposed model is interesting in the cell biological community. Addressing the following points would strengthen this paper.

Specific comments

1. The authors showed that expression of the PM-localized zDHHC11 shifts Gαo-Nt7-GFP from the Golgi to the PM, whereas expression of the Golgi-localized zDHHC3 or 7 shifts MGNC-Nt7-GFP from the PM to the Golgi, suggesting the relative expression levels of zDHHCs determine the substrate localization. To further demonstrate this model, knockdown or knockout experiment of zDHHC11 and zDHHC3/7 should be done.
2. The Gαo localization could be determined by its local S-palmitoylation level. Given that Gα proteins undergo the palmitoylation/depalmitoylation cycle, the effect of depalmitoylases (such as APTs/Lyplase) should be included as well. Experiments using inhibitors (or knockdown) could be helpful. Also, palmitate cycling on Gαo (or Gαo-Nt7) should be addressed by the pulse chase with labeled palmitate.
3. Using the SpONM assay, the authors screened the candidate zDHHCs for Gαo-Nt7 proteins. To confirm this result, the actual, specific palmitoylation of Gαo-Nt7 and wild-type Gαo proteins by zDHHC3, 7 and 11 should be investigated by biochemical approaches, such as click chemistry with 17-ODYA (17-Octadecynoic Acid).
4. Because some zDHHC PATs function together with co-factor proteins (GCP16 for zDHHC9, Golga7b for zDHHC5), the authors may want to do the SpONM assay in the presence of such cofactors.

Reviewer #2 (Remarks to the Author):

In this manuscript the authors use an imaging based strategy, using fluorescent fusions of the N-terminus of the Galpha heteromeric G protein subunit, to address the role of lipidation in subcellular localisation. The overarching question is important for the field and their key conclusion is that the position and sequence surrounding the S-acylated cysteine residues defines substrate selectivity for zDHHCs and that the subcellular localisation of Galpha is thus dictated by the localisation of cognate zDHHCs. The authors use elegant re-targeting approaches to define putative cognate zDHHCs and identify the plasma membrane targeted zDHHC 11 and predominantly Golgi localised zDHHCs 3 & 7 as recognising small variations in the position of cysteine residues in the N-terminus of Galpha.

However, there are a number of major limitations of the current work, in particular with a lack of validation of lipidation in any assay and in part as the manuscript often reads more as a collection of related experiments rather than a logical and systematic set of experiments to address this important question and so detracts from key findings.

Major issues:

1) Throughout the manuscript the authors make major assumptions about the lipidation of their constructs as there is absolutely no biochemical validation of myristoylation or S-palmitoylation with any of the various constructs/mutations. The extent to which manipulations/mutations affect lipidation is essential for interpretation throughout and statements about substrate specificity when none have been tested as S-acylation substrates is not warranted. As just a couple of many examples, in Figure 1 positional mutations that are 'predicted' to affect or have no effect on either myristoylation and/or S-acylation are assumed not tested; in Figure 2 whether any of the sequence variants, including those with two cysteines in Nt7 are S-acylated; overexpression of zDHHCs must be assumed to be increasing the already basally S-acylated pool of fusion protein but the extent this is the case is not clear. While the authors attempt in a limited number of assays to test for S-acylation the use of 2-bromopalmitate as a 'specific inhibitor of S-palmitoylation' as stated is not justified and caution has to be taken when using DHHS mutations of different zDHHCs as although in general DHHS mutants display significantly reduced S-acylation it is not always complete and needs to be validated.

2) In general most assays are based on ratios of signals in different compartments and ensuring localisation is not a function of expression is important – lipidation in many systems is a very important determinant of protein expression. Authors need to validate that manipulations/mutations do not affect expression of the various fusions (in particular when overexpressing zDHHCs) and also show quantified raw data for key experiments not simply ratios

3) A simple prediction of these assays, that is not tested, is that knockdown of endogenous zDHHC11 would reduce PM targeting of Galpha (or the Nt7-fusion) Conversely, knockdown of zDHHC3 and 7 should reduce Golgi-localisation of the MGNC-Nt7 fusion. This would greatly add to the strength of the MS.

4) Throughout manuscript authors state that their data overturns previous dogma that peripheral membrane proteins are exclusively S-acylated at the Golgi and then traffick forward. However, we have known for more than a decade this is not exclusively the case – for example elegant work by the Fukata group showing mobile zDHHC2 regulation of PSD-95 on vesicles in dendritic spines.

5)

Other points:

1) Figure 2D: Quantification of effect of 2-bromopalmitate is required

2) The section on evolution of Galpha N-terminus, apart from introducing that additional cysteine positions apart from position 3 exist does not really add to manuscript as this information, while interesting, isnt developed or really adds to the analysis

3) Why is reverse dimerization assay performed in HELA not N2a cells – is endogenous zDHHC and/or localization of overexpressed zDHHCs the same as in N2a cells?

4) Figure 4: Why is zDHHC3 or 7 relatively poor at recruiting MGNC-Nt7 to ONM whereas they are efficient at recruiting to Golgi. Are KASH constructs less catalytically active? Do these constructs bind to the Galpha fusion proteins – ie can they co-IP?

5) Authors have used a lipid overlay assay to provide evidence that differential localisation is not due to preferential lipid binding. However, this does not exclude other targeting mechanisms through interactions such as membrane microdomains or other proteins.

6) Figure 4 e to i: Overexpression of zDHHC11 with Nt7 fusion increases plasma membrane expression – however what effect does zDHHC 3 & 7 have – model would predict this would be Golgi localised? Conversely, overexpression of zDHHC3 & 7 with MGNC-Nt7 increases Golgi localisation but does overexpression of zDHHC11 increase PM localisation as would be predicted from ONM data?

7) Figure 5E&F: To fully interpret this data set the ratio for Galpha-Nt7 and MGNC-Nt7 needs to be included in bar graph.

8) The claim by the authors that they have identified “the fundamental mechanism governing protein localization” is rather overstated

9) S-acylation is reversible. The authors make no mention of whether differential location may also be a function of de-acylation of the Galpha at any location – again another key reason why biochemical evaluation of S-acylation status is required.

Reviewer #3 (Remarks to the Author):

The manuscript elaborates on the intricate regulation of G α palmitoylation, that drives its subcellular localization. The authors elegantly show the significance of coding in the N-terminus of G α as well as the subcellular location of PATs on this process. An extensive number of experiments, together with appropriate controls, have been performed. These experiments convincingly demonstrate distinct substrate specificities and sensitivities of several zDHCs for the Nt of G α and cells' ability to compartmentalize palmitoylated G α by genetically encoding the specific zDHCs. The findings reported will be an asset for the recently emerging area of cell interior GPCR-G protein signaling. Additionally, the Nt-sequence information revealed will help the grater scientific community target desired proteins to distinct subcellular regions.

Nevertheless, the following concerns should be addressed before publication.

The authors should describe how the G α palmitoylated by the outer nuclear membrane-localized zDHCs stays localized to the location of the palmitoylation. Are these cell-interior G α eventually travels to the plasma membrane?

G protein heterotrimers, even with their multiple lipid anchors in G α and G γ shuttle. They also shuttle as activated G proteins, G α GTP and G $\beta\gamma$, accessing distinct subcellular locations. What is the evolutionary advantage of then having distinct zDHCs over a universal PAT? Are the shuttling rates of G α , through the cytosol and lateral diffusion along the bound membrane, when they are localized to the plasma membrane and nuclear membrane the same?

Does G α at the nuclear membrane form heterotrimers and participate in signaling? Do they have additional PTMs to stay bound to the nuclear membrane upon palmitoylation? What prevents plasma membrane G α from reaching and staying in the outer nuclear membrane?

Additionally, minor cosmetic changes are needed for figures to make the manuscript easy to read. For instance, using gray for GFP in the left and green in the middle and again gray in the magnified images are confusing. Additionally, labels are not easy to follow. Further, given the number of conditions tested, it is highly recommended to use titles for each image subsection. For instance, having the subtitle in Figure 4f such as zDHC3 targets α -Nt7 to ONM would make it very easily comprehend.

Fig. 1k, second, and third column labels are similar. Even the legend does not provide enough information to figure out what the difference is. The mutant labeling in fig 5a-column five should be corrected. A differently colored letter for the mutation to indicate the change is advised.

REVIEWER COMMENTS

Reviewer #1 (Remarks to the Author):

The manuscript by Solis et al. investigated the role of “local S-palmitoylation” of Gao in its subcellular compartmentalization. The authors developed the SpONM (S-palmitoylation at the outer nuclear membrane) assay to evaluate the substrate specificity of zDHHC PATs in cells, by ectopically targeting zDHHCs to the outer nuclear membrane. Using this assay, the authors showed the distinct substrate specificity of zDHHCs and identified zDHHC3 and 7 as the Golgi-localized PATs and zDHHC11 as the plasma membrane (PM)-localized PAT for Gao-Nt7 proteins. Then, they showed that the expression level of the local zDHHCs PATs (Golgi vs PM) determines relative contents/distributions of Gao between Golgi and PM. Their proposed model is interesting in the cell biological community. Addressing the following points would strengthen this paper.

We would like to thank the Reviewer for his/her valuable comments and suggestions, which we address below in full. Of note, we replaced the acronym “SpONM” of our assay, mainly because it is unpronounceable. We ‘rebranded’ it as the “SwissKASH” assay in the revision.

Specific comments

1. The authors showed that expression of the PM-localized zDHHC11 shifts Gao-Nt7-GFP from the Golgi to the PM, whereas expression of the Golgi-localized zDHHC3 or 7 shifts MGNC-Nt7-GFP from the PM to the Golgi, suggesting the relative expression levels of zDHHCs determine the substrate localization. To further demonstrate this model, knockdown or knockout experiment of zDHHC11 and zDHHC3/7 should be done.

In the revised version, we analyzed the localization of Gao-Nt7 and MGNC-Nt7 upon the siRNA-mediated downregulation of zDHHCs in HeLa cells (see Fig. 8). We used HeLa – and not the murine N2a – because the siRNAs against human zDHHCs were previously validated using this cell line (Lakkaraju, et al. EMBOJ 2012) by the Lab of Gisou van der Goot (now a co-author in this study). In agreement with our

predictions, the simultaneous downregulation of the Golgi zDHHC3 and 7 resulted in a lower Golgi (and increased PM) localization of G α -Nt7, whereas MGNC-Nt7 showed a higher Golgi localization by the simultaneous depletion of the PM zDHHC5 and 11. The addition of zDHHC5 was due to new data coming from the inclusion of analysis of accessory proteins recommended by the Reviewer (see below).

2. The G α localization could be determined by its local S-palmitoylation level. Given that G α proteins undergo the palmitoylation/depalmitoylation cycle, the effect of depalmitoylases (such as APTs/Lyplase) should be included as well. Experiments using inhibitors (or knockdown) could be helpful. Also, palmitate cycling on G α (or G α -Nt7) should be addressed by the pulse chase with labeled palmitate.

We added now an analysis of the localization of G α -Nt7 and MGNC-Nt7 in N2a cells under the APT inhibitor Palmostatin B treatment (Fig. 2I-n, Supp Fig 2h-k, and Movie 1 and 2). Specifically, we performed live imaging of the cells and saw that, as expected, the localization of both constructs changed upon time upon addition of the drug. However, we were not able to see significant changes in the PM content of the constructs, indicating that depalmitoylation is not at the base of their distinct steady-state localizations. We did observe, however, significant changes in the way G α -Nt7 localizes at the Golgi region; its sharp Golgi localization became “blurry” and its co-localization with the Golgi marker diminished upon time. Based on this analysis, we consider that a pulse-chase experiment will not further improve our understanding of the different localizations of the two Nt7 constructs.

3. Using the SpONM assay, the authors screened the candidate zDHHCs for G α -Nt7 proteins. To confirm this result, the actual, specific palmitoylation of G α -Nt7 and wild-type G α proteins by zDHHC3, 7 and 11 should be investigated by biochemical approaches, such as click chemistry with 17-ODYA (17-Octadecynoic Acid).

That zDHHC3 and 7 S-palmitoylate G α subunits, such as the G α 's close relative G α i2, was biochemically demonstrated by Fukata's lab (Tsutsumi, et al. MCB 2009). However, no ability of zDHHC11 to S-palmitoylate G α subunits was observed in this paper. We provide in the Discussion an explanation why this previous study might

have failed in detecting zDHHC11's activity (page 25). Importantly, we have applied our SwissKASH assay to Gai2, observing that Gai2 is recruited to the ONM by zDHHC11 as much as Gao (see Fig below; this data is provided here for the Reviewer's attention; in the revision of our paper, we refer to this data in Discussion as 'data not shown', p.25).

Biochemical approaches have helped enormously in advancing the field of protein lipidation, but their limitations stimulated us to develop the SwissKASH assay. With this assay, we show that Gao-Nt7 and Gao are relocalized through the enzymatic activity of specific zDHHCs. We hope the Reviewer will agree with the power of these observations, enhanced in the revision, and the conclusions we drive from them.

4. Because some zDHHC PATs function together with co-factor proteins (GCP16 for zDHHC9, Golga7b for zDHHC5), the authors may want to do the SpONM assay in the presence of such cofactors.

Thanks to this suggestion of the reviewer, we were able to expand our SwissKASH assay by adding the co-factors GCP16 and Golga7b. In particular, we now show a striking specificity of the complex formed by zDHHC5 and Golga7b in the recruitment of MGNC-NT7 but not Gao-Nt7 to the ONM (see Fig. 6a-d). This effect was not observed by the closely related zDHHC5-GCP16 complex (see Supp Fig. 7a-d), although we detected a strong ONM accumulation of GCP16 by zDHHC5 (not observed by its DHHS mutant (see Supp Fig. 7e-g)). These new data helped us to

update our model (see the modified Fig. 10) adding the PM-associated zDHHC5-Golga7b complex as a key determinant for the distinct subcellular localization of Gα_{Nt7} and MGNC-NT7.

Reviewer #2 (Remarks to the Author):

In this manuscript the authors use an imaging based strategy, using fluorescent fusions of the N-terminus of the Galpha heteromeric G protein subunit, to address the role of lipidation in subcellular localisation. The overarching question is important for the field and their key conclusion is that the position and sequence surrounding the S-acylated cysteine residues defines substrate selectivity for zDHHCs and that the subcellular localisation of Galpha is thus dictated by the localisation of cognate zDHHCs. The authors use elegant re-targeting approaches to define putative cognate zDHHCs and identify the plasma membrane targeted zDHHC 11 and predominantly Golgi localised zDHHCs 3 & 7 as recognising small variations in the position of cysteine residues in the N-terminus of Galpha.

However, there are a number of major limitations of the current work, in particular with a lack of validation of lipidation in any assay and in part as the manuscript often reads more as a collection of related experiments rather than a logical and systematic set of experiments to address this important question and so detracts from key findings.

We thank the Reviewer for the comments/suggestions given to our study. Below, we address these comments in full. We have also carefully revised the information flow in our manuscript to make the data presentation more logical and straightforward, addressing the general comment of the Reviewer.

Major issues:

1) Throughout the manuscript the authors make major assumptions about the lipidation of their constructs as there is absolutely no biochemical validation of myristoylation or S-palmitoylation with any of the various constructs/mutations. The extent to which manipulations/mutations affect lipidation is essential for interpretation throughout and statements about substrate specificity when none have been tested as S-acylation substrates is not warranted. As just a couple of many examples, in Figure 1 positional mutations that are 'predicted' to affect or have no effect on either myristoylation and/or S-acylation are assumed not tested; in Figure 2 whether any of the sequence variants, including those with two cysteines in Nt7 are S-acylated; overexpression of zDHHCs must be assumed to be increasing the already basally S-

acylated pool of fusion protein but the extent this is the case is not clear. While the authors attempt in a limited number of assays to test for S-acylation the use of 2-bromopalmitate as a ‘specific inhibitor of S-palmitoylation’ as stated is not justified and caution has to be taken when using DHHS mutations of different zDHHCs as although in general DHHS mutants display significantly reduced S-acylation it is not always complete and needs to be validated.

In the revised version of this manuscript, we added biochemical evidence for the S-palmitoylation of key constructs used in our work (see Supp Fig 1m). However, we would like to mention the background knowledge regarding Gao lipidations that was not well explained in the previous version. That Gao is myristoylated and palmitoylated exclusively at Gly2 and Cys3, respectively, was biochemically demonstrated in the early 90s (Mumby, *et al*, PNAS 1990; Parenti, *et al*. Biochem J 1993, and Grassie, *et al*. Biochem J 1994; a sentence citing these references was added to the revised version). Based on this evidence, the fact that constructs derived from Gao N-terminus (WT and mutants) are myristoylated and/or palmitoylated can be inferred from their subcellular localizations, and by their presence in fractionated membrane/cytosol samples (the latter also demonstrated in the studies mentioned above). Additionally, we would like to stress that the specificity of the N-myristoylation inhibitor DDD85646 was recently confirmed in an in-depth study by Kallemeijn, *et al* (Cell Chem Biol, 2019; cited in our manuscript). And as we can judge from the available literature, the use of 2-bromopalmitate in conjunction with DHHS mutants seems widely accepted as evidence of S-palmitoylation. However, we understand the concerns of the reviewer and thus performed metabolic labeling with [³H]palmitate of Gao-Nt7 wild-type and some mutants, including MGNC-Nt7-GFP.

2) In general most assays are based on ratios of signals in different compartments and ensuring localisation is not a function of expression is important – lipidation in many systems is a very important determinant of protein expression. Authors need to validate that manipulations/mutations do not affect expression of the various fusions (in particular when overexpressing zDHHCs) and also show quantified raw data for key experiments not simply ratios

We thank the Reviewer for asking for this important control. In the revision, we added Western blot analyses for the overall expression levels of the constructs used in this study (see Supp Figs. 1g,h,k,l,n,o; 2d,e; and 8c.f). We further provide a statement to each piece of the experimental data that the effects we describe are not related to changes in the protein expression levels, which do not vary significantly for any experimental manipulation.

Regarding the use of raw data instead of ratio values for localization analysis: we used ratio values to quantify PM and Golgi as well as ONM localizations due to the large variation in the expression of the GFP-constructs within the cell population. Accordingly, each confocal image in this study was taken under optimized signal-to-noise ratio to obtain the best possible image independently of the expression level of the constructs. Thus, presenting the raw values we obtained will be misleading as the images were not taken using the same settings. To directly address this comment of the Reviewer, however, we have now performed an analysis of the PM and Golgi localization of two key constructs – Gao-Nt7-GFP and Gao-Nt31-GFP – from new confocal images recorded using the same setting at this time. As expected from our own observations, to a similar Golgi content of both constructs (mean fluorescence intensity; a.u., arbitrary units), Gao-Nt31-GFP showed consistently higher PM values compared to Gao-Nt7-GFP (see scatter plot below; $n > 50$ cells per construct). When taken all values together, we observed significant differences between the Golgi and PM content of both constructs (see bar graph below; mean \pm SEM; Student's t-test), in agreement with the data obtained using ratio values (see Fig. 1e,f). Thus, the steady-state localization patterns of these do not depend on their expression levels. These data are provided in this point-by-point response document only and are not part of the manuscript. However, we put these considerations as 'data not shown' into the revised M&M section (p.31).

3) A simple prediction of these assays, that is not tested, is that knockdown of endogenous zDHHC11 would reduce PM targeting of Galpha (or the Nt7-fusion) Conversely, knockdown of zDHHC3 and 7 should reduce Golgi-localisation of the MGNC-Nt7 fusion. This would greatly add to the strength of the MS.

We agree and thank the Reviewer for this suggestion. In the revised version, we included a loss-of-function study of the key zDHHCs and, as predicted by us and the Reviewer, we observed significant changes in the localization of Gao-Nt7 and MGNC-Nt7 (see Fig. 8a-g). Specifically, we employed siRNAs against zDHHCs

previously validated in HeLa cells (see also above comments of Reviewer #1, point #1), and observed a strong reduction in the Golgi localization of Gao-Nt7 upon simultaneous downregulation of zDHHC3 and 7, and a higher Golgi localization of MGNC-Nt7 by the siRNA against zDHHC5 and 11.

4) Throughout manuscript authors state that their data overturns previous dogma that peripheral membrane proteins are exclusively S-acylated at the Golgi and then traffick forward. However, we have known for more than a decade this is not exclusively the case – for example elegant work by the Fukata group showing mobile zDHHC2 regulation of PSD-95 on vesicles in dendritic spines.

We are well aware of Fukata's work on PSD-95. Indeed, we cited this paper on zDHHC2 in two sections of the manuscript: when we introduced the "SwissKASH" system (page 14 in the revision) and in Discussion as one of the few reports of local S-palmitoylation (page 23 in revision). By taking the available literature as a whole, however, we observed a certain dominance of the "Golgi only" model for the S-palmitoylation of PMPs, mainly based on Bastiaens' Lab work. We also recognized that the "Golgi only" view is been currently challenged, as more evidence for local S-palmitoylation accumulates. In fact, we explicitly described these opposite views in the introduction section (page 4 in revision), citing the Fukata's pivotal review (Biochem Soc Trans 2015) and a recent review by Philippe and Jenkins (Mol Membr Biol 2020). Nevertheless, in this revised version we toned down the sentences on the "Golgi only" model throughout the paper.

Other points:

1) Figure 2D: Quantification of effect of 2-bromopalmitate is required

Due to the addition of new data we moved the image in Figure 2D to the Supplementary Information (Supp Fig. 2c). As mentioned in response to the point #1 of this Reviewer, we biochemically demonstrated the S-palmitoylation of MGNC-Nt7-GFP (Supp Fig. 1m). The strong and consistent microscopy-viewed change in the localization of MGNC-Nt7-GFP induced by 2-BrPal (cf. Fig. 2a and Supp Fig. 2c),

combined with this biochemical data, makes adding another quantification to this highly data-charged paper excessive, in our opinion.

2) The section on evolution of Galpha N-terminus, apart from introducing that additional cysteine positions apart from position 3 exist does not really add to manuscript as this information, while interesting, isn't developed or really adds to the analysis

We politely disagree with the Reviewer regarding the lack of overall relevance of the evolution section of the manuscript. Our main goal was to deconstruct the key component(s) governing Gao PM and Golgi localization, underlining its importance by the evolutionary conservation. Nevertheless, we do understand that the evolution section might break the flow of the manuscript to some extent. Thus, we took this section out of the main text and moved it into the Supplementary Information as a whole (as Supplementary Methods, Supplementary Notes, and Notes Figure 1). We also removed a sentence from Discussion that referred to the evolution analysis.

3) Why is reverse dimerization assay performed in HELA not N2a cells – is endogenous zDHHC and/or localization of overexpressed zDHHCs the same as in N2a cells?

We show now that the localization patterns of HA-tagged zDHHC3, 7, and 11 in HeLa cells (see Supp Fig 9a-c) are the same as in N2a cells. The PM localization of zDHHC5 has been extensively reported, and we also detected RFP-zDHHC5 at the PM in HeLa cells (Fig 4c).

The choice of HeLa cells for the RD assay has been dictated by the much better suitability of these large cells, as we wrote in the main text: “for better visualization”. In fact, we initially developed this assay in N2a cells (see the figure below), and moved to HeLa cells due to their much bigger cell body. N2a cells have a roundish cell body of 12-17 μm in diameter, and a small cytosolic area surrounding the nucleus. Although N2a cells are ideal for PM and Golgi visualization/quantification, the signal derived from the FM4-clusters overlaps with the Golgi. On the other hand, HeLa cells are flat, reach up to 50 μm in longitude, and have a much larger cytosolic

area making them better suitable for this assay. In this revised version, we added a clearer explanation of why we used HeLa instead of N2a; we also now state that the RD assay was initially developed for N2a cells and we then moved to HeLa cells for the sake of better visualization (page 12).

4) Figure 4: Why is zDHHC3 or 7 relatively poor at recruiting MGNC-Nt7 to ONM whereas they are efficient at recruiting to Golgi. Are KASH constructs less catalytically active? Do these constructs bind to the Galpha fusion proteins – ie can they co-IP?

Precaution should be taken when comparing the apparent levels of MGNC-Nt7 at the ONM in the SwissKASH assay vs. at the Golgi by the co-expression of HA-zDHHC3/7. For instance, the GFP-signal in a confocal image of the ONM represents the amount of a construct in a thin section of a single membrane bilayer. At the Golgi, an equally thin confocal section contains a much dense region of folded membranes forming the cisternae stack. Thus, the apparent higher accumulation of MGNC-Nt7 at the Golgi vs. the ONM might rather relate to the large difference in membrane density between the two compartments.

We also added Immunoprecipitation (IP) analyses and showed that Gαo-Nt7-GFP and MGNC-Nt7-GFP did not co-IP the KASH constructs of zDHHC3, 7, 5, and 11 (Supp Figs. 4f; 5c,d; and 7h). As a positive interacting control, we showed that a SUN2-GFP construct was able to co-IP all KASH-fusions tested. An IP analysis was also done for MGSSCSR-Nt7 and MGLLCSR-Nt7, and both showed no co-IP of mRFP-zDHHC11-KASH (Supp Fig. 6d). As an additional positive control (not shown in the manuscript), we performed an IP against GFP-SNAP23 and Gαo-Nt7-GFP, and detected the co-precipitation of the KASH-construct of zDHHC17 only by SNAP23 (see image below). SNAP23 is known to interact directly with the ankyrin

repeat domain in zDHHC17. Thus, these new data indicate that the different accumulation of the Nt7 constructs at the ONM is not due to differential protein-protein interactions with the different zDHHC-KASH fusions.

5) Authors have used a lipid overlay assay to provide evidence that differential localisation is not due to preferential lipid binding. However, this does not exclude other targeting mechanisms through interactions such as membrane microdomains or other proteins.

We agree with the reviewer that our study cannot fully exclude the existence of additional targeting mechanism(s) for Gao-Nt7 and MGNC-Nt7 to the PM and/or Golgi. However, the fact that both constructs can be redirected to the ONM (and surrounding ER) by the KASH-fusion of zDHHC11 speaks in favor of local S-palmitoylation, as specific PM/Golgi microdomains and interacting partners are not expected at the ER/ONM. That the PM/Golgi targeting of the Nt7 constructs can be efficiently modified by manipulating the expression levels of specific zDHHCs also supports local S-palmitoylation as the major player for their compartmentalization. Therefore, we believe that additional unknown factors do not play essential roles in the steady-state localization of Gao-Nt7 and MGNC-Nt7, but may represent auxiliary regulatory functions.

6) Figure 4 e to i: Overexpression of zDHHC11 with Nt7 fusion increases plasma membrane expression – however what effect does zDHHC 3 & 7 have – model would predict this would be Golgi localised? Conversely, overexpression of zDHHC3 & 7 with MGNC-Nt7 increases Golgi localisation but does overexpression of zDHHC11 increase PM localisation as would be predicted from ONM data?

The 'starting' localization of G α -Nt7-GFP is predominantly at the Golgi, and strong PM relocalization of G α -Nt7-GFP is observed upon overexpression of the PM-localized zDHHC11 (Fig. 7a,b). Overexpression of the Golgi-localizing zDHHC3/7 cannot further aggravate the already predominant Golgi localization of G α -Nt7-GFP (Supp Fig. 8g,h). In contrast, our new data show that downregulation of the Golgi zDHHCs dramatically reduces the Golgi localization and increases the PM localization of G α -Nt7-GFP (Fig. 8a,b).

Reciprocal findings are presented for the MGNC-Nt7-GFP construct: it's starting almost exclusive localization at PM and endosomes is not influenced further by overexpression of the PM-localizing zDHHC11 (Supp Fig. 8i), but is strongly affected by depletion of PM zDHHCs (new Fig. 8f) or by overexpression of the Golgi zDHHCs (Fig. 7e-j).

7) Figure 5E&F: To fully interpret this data set the ratio for Galpha-Nt7 and MGNC-Nt7 needs to be included in bar graph.

We now added the control values to the graphs (Fig. 7c,d,g,h,k,l) and corresponding representative images (Sup Fig. 8a,b).

8) The claim by the authors that they have identified “the fundamental mechanism governing protein localization” is rather overstated

We have toned down our statements throughout the text.

9) S-acylation is reversible. The authors make no mention of whether differential location may also be a function of de-acylation of the Galpha at any location – again another key reason why biochemical evaluation of S-acylation status is required.

We added now an analysis of the localization of G α -Nt7 and MGNC-Nt7 in N2a cells under the APT inhibitor Palmostatin B (Fig. 2l-n, Supp Fig 2h-k, and Movie 1 and 2). Specifically, we performed live imaging of the cells and saw that, as expected, the localization of both constructs changed upon time upon addition of the drug. However, we were not able to see significant changes in the PM content of the constructs, indicating that depalmitoylation is not at the base of their distinct steady-state localizations. We did observe, however, significant changes in the way G α -Nt7 localizes at the Golgi region; its sharp Golgi localization became “blurry” and its co-localization with the Golgi marker diminished upon time. Based on this analysis, we consider that a pulse-chase experiment will not further improve our understanding of the different localizations of the two Nt7 constructs.

Reviewer #3 (Remarks to the Author):

The manuscript elaborates on the intricate regulation of Gαo palmitoylation, that drives its subcellular localization. The authors elegantly show the significance of coding in the N-terminus of Gα as well as the subcellular location of PATs on this process. An extensive number of experiments, together with appropriate controls, have been performed. These experiments convincingly demonstrate distinct substrate specificities and sensitivities of several zDHHs for the Nt of Gα and cells' ability to compartmentalize palmitoylated Gα by genetically encoding the specific zDDHCs. The findings reported will be an asset for the recently emerging area of cell interior GPCR-G protein signaling. Additionally, the Nt-sequence information revealed will help the greater scientific community target desired proteins to distinct subcellular regions.

We thank the Reviewer for the valuable comments/suggestions. We address in detail his/her comments in full below.

Nevertheless, the following concerns should be addressed before publication. The authors should describe how the Gα palmitoylated by the outer nuclear membrane-localized zDDHCs stays localized to the location of the palmitoylation.

We do not think that the localization at the ONM of Gαo constructs requires much explanation. Myristoylated and/or palmitoylated peptides possess affinity for different type of lipid membranes (e.g. review by Bhatnagar and Gordon, Trends Cell Biol. 1997; 7:14), therefore, it is not surprising that our constructs are capable of association with the ONM upon local S-palmitoylation. Indeed, association with the nuclear membrane of a soluble protein via S-palmitoylation has been previously shown for Rif1 (Rap1-interacting factor 1; Park, et al. PNAS. 2011; 108:14572). Moreover, as some zDHHs showed a clear localization to the nuclear membrane (Supp Fig. 3b), S-palmitoylation at this compartment might be more frequent than currently known. Accordingly, S-palmitoylation of nuclear proteins have been detected on multiple screenings; information can be found in the SwissPalm database (<https://swisspalm.org/>).

Are these cell-interior G α eventually travels to the plasma membrane?

S-palmitoylation is a reversible process, therefore, the possibility that G α constructs targeting the ONM might eventually be depalmitoylated and palmitoylated back at the PM exists. Also, we would like to point out that the targeting of G α constructs by the KASH-fusions is not exclusively at the ONM, but also the ER (e.g. Fig. 5b,d). Thus, G α constructs might “jump” from the ER to the PM at the ER-PM Contact Sites. However, as the ONM localization of G α constructs is not physiological, we believe that the further study of their trafficking from the ONM is not the scope of this work.

G protein heterotrimers, even with their multiple lipid anchors in G α and G γ shuttle. They also shuttle as activated G proteins, G α GTP and G $\beta\gamma$, accessing distinct subcellular locations. What is the evolutionary advantage of then having distinct zDHCs over a universal PAT?

This is a very interesting point. The Reviewer is right pointing out that shuttling of G $\beta\gamma$ heterodimers to distinct subcellular localizations upon GPCR activation is a well-studied phenomenon. However, this is not the case for G α which stays at the PM upon activation (Akgoz, et al. JBC. 2004; 279:51541, and our own work in Solis, et al. Cell. 2017; 170:939). In Discussion, we wrote regarding the many zDHCs (Page 24): “Another interesting implication of this model is that cells and tissues could shift the localization – and associated function – of PMPs toward one specific compartment or another by controlling the expression of zDHCs”. As PMPs are constantly depalmitoylated, a rapid activation/inactivation of specific zDHCs at a specific compartment might also regulate PMPs’ local S-palmitoylation and function in an acute manner. To sum up, we believe that multiple zDHCs with different localization patterns permit achieving easily tunable regulation of localization and hence activity of the many PMPs.

An additional advantage of having multiple PATs is that mutations on a universal enzyme might have devastating effects for the cell/tissue/organism as a large number of palmitoylated proteins will be affected.

Are the shuttling rates of G α , through the cytosol and lateral diffusion along the bound membrane, when they are localized to the plasma membrane and nuclear membrane the same?

These are also interesting points. However, since the ONM localization of G α constructs is not of physiological relevance, we do not think that the analysis of their dynamics at the ONM is within the scope of this work.

Does G α at the nuclear membrane form heterotrimers and participate in signaling?

We are glad that the Reviewer is clearly intrigued by the artificial localization of G α , as can be judged by this set of questions. We also wondered whether the relocalized G protein maintained its signaling competence in this new location. In order to address this, we performed the SwissKASH assay using the full-length G α -GFP with the co-expression of G β 1 γ 3 to test the recruitment of the heterotrimeric complex, and RGS19 as a signaling molecule downstream of G α . In fact, we now show the co-recruitment of G β 1 γ 3 to the ONM (as well as the ER) by G α -GFP (Fig. 9g). Conversely, RGS19 was not co-recruited to the ONM by G α -GFP (Supp Fig 10f), as G α is most likely expressed in its inactive GDP-bound state. Thus, we used the constitutive active Q205L mutant of G α – which is incapable of GTP hydrolysis – and observed a prominent co-recruitment of RGS19 at the ONM (Fig. 9h).

Do they have additional PTMs to stay bound to the nuclear membrane upon palmitoylation?

This issue is related to the first point of the Reviewer. As we mentioned above, the association of myristoylated and/or palmitoylated peptides with the ER/ONM membranes should not require any additional modification other than lipidations. Additionally, the short sequence of the Nt7 constructs does not contain additional residues expected to undergo PTM that could influence association with membranes.

What prevents plasma membrane G α from reaching and staying in the outer nuclear membrane?

We believe that our work answered this question. Under normal circumstances, Gao appears not to be efficiently recognized by the zDHHCs residing at the ONM (and ER), and is therefore mainly palmitoylated at the PM and Golgi where it resultantly resides. In order to address this point directly, we co-expressed Gao-GFP together with the zDHHCs resident at the ONM/ER (Supp Fig. 9e). As we saw no mislocalization of Gao-GFP, we concluded that Gao might not to be a good substrate for these zDHHCs. When we expressed the ER zDHHCs in HeLa cells, we noted that two of them targeted the PM as well, a pattern that was not seen in N2a cells (Supp Fig. 3b vs. 9e). Thus, we included them in our SwissKASH assay but found that they did not recruit Gao-Nt7-GFP or MGNC-Nt7-GFP to the ONM (Supp Fig. 10a-d).

Additionally, minor cosmetic changes are needed for figures to make the manuscript easy to read. For instance, using gray for GFP in the left and green in the middle and again gray in the magnified images are confusing. Additionally, labels are not easy to follow. Further, given the number of conditions tested, it is highly recommended to use titles for each image subsection. For instance, having the subtitle in Figure 4f such as zDHHC3 targets α -Nt7 to ONM would make it very easily comprehend.

We fully understand that a composite of separate panels for each channel might be best presentation format for confocal images, and that is the format we used in all Supplementary Figures. In the main Figures, however, we presented the images in a different format for several reasons: (i) Due to space limitations. (ii) We used a gray image for GFP to the left of the merged image, because the GFP channel contains the most relevant information, and (iii) gray because the green color on a black background is harder to visualize, especially when printed on paper (this applies also for the gray in the magnifications). (iv) In most of the main images we show the red channel only in the merge, because it contains the Golgi marker or a KASH-fusion, that do not necessarily need a separate image (except in magnifications). In other images, however, we show all channels separated, as they contain relevant information by their own (e.g., IF of Golga7b). We now added a detailed description in each figure legend explaining the format we used. Subtitles for legends are not standard for Nat Commun, therefore we increased the number of main and supplementary figures but kept the legends as concise as possible.

Fig. 1k, second, and third column labels are similar. Even the legend does not provide enough information to figure out what the difference is. The mutant labeling in fig 5a-column five should be corrected. A differently colored letter for the mutation to indicate the change is advised.

We thank the Reviewer for noticing these oversights. In Fig 1k (now Fig. 1o), the labeling was corrected, we meant to write S6C and S6G instead of two times S6C. We also corrected the wrong labeling in Fig 5a (now Fig. 5o).

We used two different ways to describe mutations in Gao: one that addresses point mutations, keeping “Gao” in the name such as “Gao-S6C”, and a second one that described mainly artificial sequences that differ from Gao-Nt7 by at least 2 residues such as “MGNC”. We believe that the format we used for point mutations is standard and self-explanatory. We agree that the second format might be confusing, and we underlined in the figures the letters/residues that deviate from Gao wild-type sequence, e.g., “MGNC” (explained in figure legends). We prefer the underlined letter instead of colored letters, because the construct names are already colored in many images. Once again, many thanks for helping us to improve the clarity of our data presentation.

Reviewers' comments:

Reviewer #1 (Remarks to the Author):

The authors have appropriately addressed my concerns.

Reviewer #2 (Remarks to the Author):

The authors have made some significant improvements to their manuscript and adequately addressed some of the concerns raised in the original critique.

However a number of major issues remain, including most importantly that they have not really addressed the central concern whether key manipulations (zDHHC KD, zDHHC over expression, pharmacological inhibition of zDHHC or acylthioesterase etc) in fact change S-acylation of fusion constructs. This is a real pity as the manuscript has the potential to provide new insights into Galpha trafficking

1) The authors have now included (Suppler Fig1m) a 3H-palmitate assay revealing S-acylation of some key constructs. As the authors have now better highlighted in the MS we already know alpha is S-acylated and this is not the issue raised. The key point that they have not defined whether this changes with any of the key manipulations is a concern and detracts from the rigor of the manuscript. This is particularly important if they want to claim selectivity/specificity for zDHHCs in these mechanism for controlling S-acylation of Galpha

a) In Fig S1M the C3N construct is still labelled by 3H-palmitate that is sensitive to hydroxylamine. This is a concern as this suggests in their fusion construct an additional S-acylated cysteine is present that may affect interpretation. In the results the authors state this is likely in GFP and is a 'common' finding in the field. Many studies in field have used GFP fusions without this problem and so the authors should state the evidence that GFP is 'commonly' S-acylated and discuss the implications for their analysis.

b) Although they do not test S-acylation the authors still make assumptions base don imaging assay. For example, in response to analysis of role of deacylation they have used Palmostatin B that inhibits some acylthioesterases. They conclude that as no change in PM location is observed (while see local fuzzy change in Golgi location) this supports that deacylation is not a key mechanism. Without showing that Palm B in fact changes S-acylation of Galpha constructs this cannot be concluded.

2) The data reported as 'not shown' on p 31 that addresses the important issue that manipulations don't affect total construct expression should be included in MS - at least authors should include the bar chart included in response to reviewers as supplementary data.

3) In the absence of validation that BrPal changes S-acylation of constructs this data should at least be quantified as previously suggested for Fig2a and SFig2c). Such quantification I would concur is not excessive or unreasonable.

Reviewer #3 (Remarks to the Author):

The authors have addressed my comments and concerns sufficiently.

Reviewers' comments:

Reviewer #1 (Remarks to the Author):

The authors have appropriately addressed my concerns.

We genuinely thank the Reviewer for helping us significantly improve this manuscript.

Reviewer #2 (Remarks to the Author):

The authors have made some significant improvements to their manuscript and adequately addressed some of the concerns raised in the original critique.

However a number of major issues remain, including most importantly that they have not really addressed the central concern whether key manipulations (zDHHC KD, zDHHC over expression, pharmacological inhibition of zDHHC or acylthioesterase etc) in fact change S-acylation of fusion constructs. This is a real pity as the manuscript has the potential to provide new insights into Galpha trafficking.

We thank the Reviewer for this insistence in requiring the biochemical confirmation of our observations. Following these requirements, we have now performed a number of additional experiments, that fully address the Reviewers' concerns, as detailed below. We hope now the Reviewer will agree that the paper should be published and will provide insights into Galpha trafficking, important for the whole field.

1) The authors have now included (Suppl Fig1m) a 3H-palmitate assay revealing S-acylation of some key constructs. As the authors have now better highlighted in the MS we already know alpha is S-acylated and this is not the issue raised. The key point that they have not defined whether this changes with any of the key manipulations is a concern and detracts from the rigor of the manuscript. This is particularly important if they want to claim selectivity/specificity for zDHHCs in these mechanism for controlling S-acylation of Galpha

In this revised version, we now include additional quantitative data from several 3H-palmitate metabolic labelling experiments for the most relevant constructs, under the conditions asked by the Reviewer. Particularly, we show that 2-bromopalmitate strongly impairs, while Palmostatin B – increases 3H-palmitate incorporation in several constructs (new Fig 1o-q and Supp Fig 1o-q). We also show that incorporation of 3H-palmitate in the two most relevant constructs (Gao-Nt7-GFP and MGNC-Nt7-GFP) is differentially regulated by the overexpression/downregulation of the Golgi- and PM-associated zDHHCs (new Fig 7m-o and Fig 8h,i). These new results confirm our main conclusion that these two closely related Nt7 sequences are differentially localized in cells due to the local and specific action of the resident zDHHCs.

a) In Fig S1M the C3N construct is still labelled by 3H-palmitate that is sensitive to hydroxylamine. This is a concern as this suggests in their fusion construct an additional S-acylated cysteine is present that may affect interpretation. In the results the authors state this is likely in GFP and is a 'common' finding in the field. Many studies in field have used

GFP fusions without this problem and so the authors should state the evidence that GFP is 'commonly' S-acylated and discuss the implications for their analysis.

We politely disagree with the Reviewer's opinion that S-acylation of GFP "is a concern...that may affect interpretation." It could be of concern if considered alone, but not at all of concern when considered as part of the whole body of the data we presented (and now expanded in the revision). For instance, fractionation experiments show that the G α -C3N-Nt7 construct is poorly associated to membranes (~5% vs ~60% for G α -Nt7; Fig 1k,l). This residual membrane binding property of G α -C3N-Nt7 appears to be exclusively due to N-myristylation of Gly2 and not to S-acylation of GFP, as the G2L mutant is virtually absent in membrane fractions (Fig 1k,l) despite the fact that both constructs are similarly labelled with 3H-palmitate (Supp Fig 1m).

We have now additionally performed 3H-palmitate labelling for the analogous full-length constructs of G α (Supp Fig 1o-q), detecting very similar levels of 3H-palmitate incorporation into the C3N mutants of G α full-length and Nt7 (~10% of the corresponding wild type constructs; Fig 1o,q and Supp Fig 1o,q). These results suggest that GFP is stochastically lipidated to a certain degree when overexpressed. In our communication with our collaborators (Lab of Gisou van der Goot), we learnt that S-acylation of GFP has been occasionally mentioned in conferences but we were unable to find any publication describing this effect. Thus, we removed this claim from our manuscript as it represents an overstatement from our side. However, this marginal S-acylation of GFP does not have any significant implication for our study, as it does not result in the membrane association of our constructs.

b) Although they do not test S-acylation the authors still make assumptions based on imaging assay. For example, in response to analysis of role of deacylation they have used Palmostatin B that inhibits some acylthioesterases. They conclude that as no change in PM location is observed (while see local fuzzy change in Golgi location) this supports that deacylation is not a key mechanism. Without showing that Palm B in fact changes S-acylation of Galpha constructs this cannot be concluded.

This point seems to represent an unfortunate misreading by the Reviewer. We did not claim that "deacylation is not a key mechanism." On the contrary, we wrote: "This analysis supports the notion that the steady-state localization of PMPs results from a palmitoylation/depalmitoylation equilibrium [Rocks. Cell 2010]." Our statement was directed to the role of Golgi-to-PM trafficking and not to deacylation: "the fact that the PM content of G α -Nt7 was not affected by Palmostatin B argues against a constant Golgi-to-PM flow of the construct." The latter statement is supported by the data derived from the reverse dimerization assay, which allowed us to directly analyze the trafficking of the constructs (Fig 3).

Further, as mentioned above, we have now included new data showing the changes Palm B induces in 3H-palmitate labelling of the constructs (new Fig 1o-q and Supp Fig 1o-q).

2) The data reported as 'not shown' on p 31 that addresses the important issue that manipulations don't affect total construct expression should be included in MS - at least authors should include the bar chart included in response to reviewers as supplementary data.

As suggested by the Reviewer, these data is now added to the revised manuscript as Supp Fig 10g,h.

3) In the absence of validation that BrPal changes S-acylation of constructs this data should at least be quantified as previously suggested for Fig2a and SFig2c). Such quantification I would concur is not excessive or unreasonable.

This point is directly related to the point #1 mentioned above (of note, SFig2c is now SFig2f). We have now included the validation and quantification of the 2-BrPal effect on the 3H-palmitate labelling of the MGNC-Nt7-GFP construct (Fig 1o,p).

Reviewer #3 (Remarks to the Author):

The authors have addressed my comments and concerns sufficiently.

We very much appreciate the feedback we received from the Reviewer, which helped us strengthen our manuscript.

REVIEWERS' COMMENTS

Reviewer #2 (Remarks to the Author):

The authors have adequately addressed my key concerns with both new data and relevant discussion as appropriate

REVIEWER COMMENTS

Reviewer #2 (Remarks to the Author):

The authors have adequately addressed my key concerns with both new data and relevant discussion as appropriate.

We truly thank the Reviewer for her/his feedback that helped us significantly improve this manuscript.